# Causal Forcing: Autoregressive Diffusion Distillation Done Right for High-Quality Real-Time Interactive Video Generation

Hongzhou Zhu [* 1 2]  Min Zhao [* 1 2]  Guande He [3]  Hang Su [1]  Chongxuan Li [4 5 6]  Jun Zhu [1 2]

## Abstract

To achieve real-time interactive video generation, current methods distill pretrained bidirectional video diffusion models into few-step autoregressive (AR) models, facing an *architectural gap* when full attention is replaced by causal attention. However, existing approaches do not bridge this gap theoretically. They initialize the AR student via ODE distillation, which requires *frame-level injectivity*, where each noisy frame must map to a unique clean frame under the PF-ODE of an *AR teacher*. Distilling an AR student from a bidirectional teacher violates this condition, preventing recovery of the teacher's flow map and instead inducing a conditional-expectation solution, which degrades performance. To address this issue, we propose *Causal Forcing*, which uses an autoregressive teacher for ODE initialization to bridge the architectural gap, and then applies the same DMD procedure as in Self Forcing. Empirical results show that our method outperforms all baselines across all metrics, surpassing the SOTA Self Forcing by 19.3% in Dynamic Degree, 8.7% in VisionReward, and 16.7% in Instruction Following. Project page: https://thu-ml.github.io/CausalForcing.github.io/; the code: https://github.com/thu-ml/Causal-Forcing.

## 1. Introduction

Recent years have witnessed rapid progress in autoregressive (AR) video diffusion models (Jin et al., 2024; Teng et al., 2025; Chen et al., 2025; Wu et al., 2025). By adopting a frame-level autoregressive formulation with diffusion within each frame, AR video diffusion enables a wide range of real-time and interactive applications, including world modeling (Mao et al., 2025; Sun et al., 2025a; Hong et al., 2025), game simulation (Ball et al., 2025; Tang et al., 2025), embodied intelligence (Feng et al., 2025), and interactive content creation (Shin et al., 2025; Huang et al., 2025b; Ki et al., 2026; Xiao et al., 2025). Despite their promise, the computational burden of multi-step diffusion sampling severely limits their real-time capabilities.

To alleviate this latency bottleneck, recent works (Huang et al., 2025a; Yin et al., 2025) distill a powerful pretrained *bidirectional* video diffusion model into a few-step *autoregressive* student model. This is typically achieved via a two-stage pipeline: an initial ODE distillation to initialize the AR student, followed by DMD (Yin et al., 2024) to further boost performance. However, compared to standard step-distillation, such AR distillation faces a more fundamental challenge beyond the shared sampling-step gap, namely, the *architectural gap*. This gap arises from converting a bidirectional model, which has access to future frames, into a causal architecture that conditions solely on past context. Empirically, we find that even when distilled from the same bidirectional teacher, SOTA AR distillation methods (Huang et al., 2025a) still lag significantly behind standard DMD, which distills a bidirectional student (see Fig. 1).

In this paper, we show that the performance degradation stems from the failure of existing methods to properly address the architectural gap theoretically (see Fig. 3 and Sec. 3.2 ). Through a controlled experiment, we first show that this gap cannot be resolved by the DMD stage and should instead be addressed during the preceding ODE initialization. Crucially, a key requirement for ODE distillation is *injectivity* (Liu et al., 2022). In standard ODE distillation that distills a bidirectional teacher into a bidirectional student, injectivity naturally holds at the video level. In contrast, for an AR student, injectivity must hold at the frame level: each noisy frame must map to a *unique* clean

---

[*]Equal contribution [1]Dept. of Comp. Sci. & Tech., BNRist Center, THU-Bosch ML Center, Tsinghua University. [2]ShengShu. [3]The University of Texas at Austin. [4]Gaoling School of Artificial Intelligence Renmin University of China Beijing, China. [5]Beijing Key Laboratory of Research on Large Models and Intelligent Governance. [6]Engineering Research Center of Next-Generation Intelligent Search and Recommendation, MOE.. Correspondence to: Jun Zhu <dcszj@tsinghua.edu.cn>.

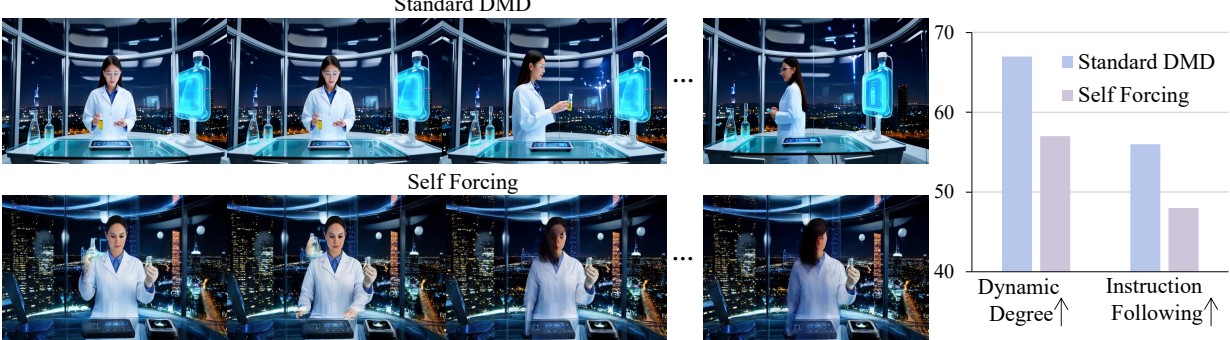

Figure 1. **Limitations of existing methods.** While distilling from the same bidirectional base model, SOTA *autoregressive* diffusion distillation methods like Self-Forcing still lag significantly behind standard DMD, which distills a *bidirectional* student.

frame under the PF-ODE of the *AR teacher*. We refer to this requirement as *frame-level injectivity*. However, existing methods (Huang et al., 2025a; Yin et al., 2025) distill an AR student directly from a bidirectional teacher, allowing the *same* noisy frame to correspond to multiple *different* clean frames. This violation of frame-level injectivity results in blurred and inconsistent video generation.

Building on the above analysis, we propose *Causal Forcing*, which bridges the architectural gap by performing *ODE distillation initialization with an AR teacher* (see Sec. 3.3). We first train an AR diffusion model using teacher forcing, where we show that diffusion forcing is inferior to teacher forcing for AR diffusion training both theoretically and empirically. With this AR diffusion model as the teacher, we then perform *causal ODE distillation* by sampling its PF-ODE trajectories and training the AR student accordingly. Crucially, since the teacher is autoregressive rather than bidirectional, its PF-ODE naturally satisfies frame-level injectivity, enabling the student to accurately learn the flow map. Finally, we apply the same DMD procedure as in Self Forcing to obtain a few-step AR student with high quality, enabling high-quality efficient real-time video generation.

To validate our approach, we conduct comprehensive evaluations against various baseline models (Wan et al., 2025; Ha-Cohen et al., 2024; Deng et al., 2024; Jin et al., 2024; Chen et al., 2025; Teng et al., 2025; Yin et al., 2025; Huang et al., 2025a). Experiments show that our method consistently outperforms all baselines across all metrics, with significant gains in dynamic degree, visual quality, and instruction-following capability. Remarkably, under the same training budget as existing distilled autoregressive video models, it surpasses the SOTA Self-Forcing (Huang et al., 2025a) baseline by 19.3% in Dynamic Degree (Huang et al., 2024), 8.7% in VisionReward (Xu et al., 2024), and 16.7% in Instruction Following, while maintaining the same inference latency, demonstrating the effectiveness of our method.

## 2. Background

### 2.1. Diffusion Models

Diffusion models (Ho et al., 2020; Song et al., 2020) gradually perturb data $\boldsymbol{x}_0 \sim p_{\text{data}}(\boldsymbol{x}_0)$ through a forward diffusion process to learn its distribution. This process follows a transitional kernel $q_{t|0}(\boldsymbol{x}_t|\boldsymbol{x}_0)$ given by $\boldsymbol{x}_t = \alpha_t \boldsymbol{x}_0 + \sigma_t \boldsymbol{\epsilon}$, $\boldsymbol{\epsilon} \sim \mathcal{N}(\mathbf{0}, \boldsymbol{I})$, $t \in [0, T]$, where $\alpha_t, \sigma_t$ are the predefined noise schedule. To match the data distribution, the model can be trained under a variety of parameterizations (Ho et al., 2020; Kingma & Gao, 2023; Salimans & Ho, 2022). A typical parameterization is flow matching (Lipman et al., 2022), which uses the velocity prediction. The model $\boldsymbol{v}_\theta$ is trained to minimize the weighted mean square error $\mathbb{E}_{\boldsymbol{x}_0, \boldsymbol{\epsilon}, t}[w(t) ||\boldsymbol{v}_\theta(\boldsymbol{x}_t, t) - \boldsymbol{v}_t||^2]$. Under a typical noise schedule (Liu et al., 2022) that $\alpha_t = 1 - t, \sigma_t = t$ and $T = 1$, $\boldsymbol{v}_t$ is defined by $\boldsymbol{v}_t := \frac{d\boldsymbol{x}_t}{dt} = \boldsymbol{\epsilon} - \boldsymbol{x}_0$. At this point, sampling can be done by solving the probability flow ordinary differential equation (PF-ODE) (Song et al., 2020)

$$\mathrm{d}\boldsymbol{x}_t = \boldsymbol{v}_\theta(\boldsymbol{x}_t, t)\mathrm{d}t, \quad \boldsymbol{x}_T \sim \mathcal{N}(\mathbf{0}, \boldsymbol{I}), \quad t : T \to 0. \quad (1)$$

### 2.2. Autoregressive Video Diffusion Models

Despite their success in video generation (Yang et al., 2024; Kong et al., 2024), full-sequence diffusion models generate all frames in a single shot, preventing user interaction. Autoregressive (AR) video diffusion models instead generate frames sequentially, aiming to model the distribution of $N$-frame videos via an autoregressive factorization[1] $p_\theta(\boldsymbol{x}_0^{1:N}) = \prod_{i=1}^{N} p_\theta(\boldsymbol{x}_0^i \mid \boldsymbol{x}_0^{<i})$, where each conditional distribution $p_\theta(\boldsymbol{x}_0^i \mid \boldsymbol{x}_0^{<i})$ is modeled by standard diffusion. This mechanism enables users to steer subsequent frames based on the generated content, thereby enabling interactivity, exemplified by Google's Genie 3 (Ball et al., 2025).

To achieve this, two typical training strategies can be adopted, namely teacher forcing (TF) (Jin et al., 2024) and

---

[1] In practice, generation is typically performed in chunks rather than frame-by-frame. We omit this detail here for simplicity.

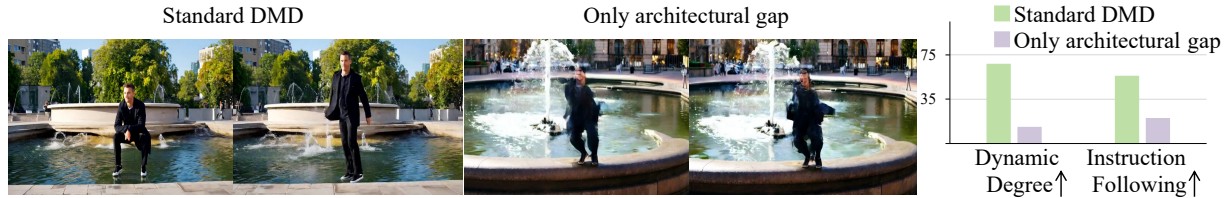

*Figure 2.* **DMD fails to bridge the architectural gap.** Initializing the autoregressive student with standard DMD removes the sampling-step gap and isolates the architectural gap, yet still underperforms standard DMD. This indicates that the architectural gap cannot be resolved by the DMD stage and should instead be addressed during the preceding ODE initialization.

diffusion forcing (DF) (Chen et al., 2024; Song et al., 2025). TF aims to learn $p_{\text{data}}(x_0^i \mid x_0^{<i})$ conditioning on a clean prefix of past frames $x_0^{<i}$. To enable this training paradigm in practice, a commonly used strategy (Teng et al., 2025) is concatenating the clean video with its noisy counterpart and applying a causal attention mask, so that $x_t^i$ can attend to $x_0^{<i}$. In contrast, DF targets the noisy-conditioned distribution $p_{\text{DF}}(x_0^i \mid x_t^{<i})$, with noise added independently to each frame via $q_{t|0}(x_t^{<i} \mid x_0^{<i})$. Rather than feeding a clean prefix as in TF, DF lets $x_t^i$ directly attend to the noisy prefix $x_t^{<i}$. See more related work in Appendix A.

### 2.3. Consistency Distillation and ODE Distillation

To enable real-time generation, multi-step diffusion models are typically distilled into few-step models. A typical approach is Consistency Distillation (CD) (Song et al., 2023; Song & Dhariwal, 2023). This is achieved by learning a flow map $G_\theta : (x_t, t) \mapsto x_0$ that maps $x_t$ to the clean endpoint $x_0$ of the teacher diffusion model's PF-ODE. Under the boundary condition $G_\theta(x, 0) \equiv x$, the model $G_\theta$ can be trained by minimizing $\mathbb{E}_{x_0, \epsilon, t}[w(t)d(G_\theta(x_t, t), G_{\theta^-}(\hat{x}_{t-\Delta t}, t - \Delta t))]$, where $x_t$ is obtained via the forward diffusion process, $\hat{x}_{t-\Delta t}$ is obtained by solving one ODE step from $x_t$ using the teacher diffusion model, $\theta^-$ is a running average of $\theta$ with stop-gradient, and $d(\cdot, \cdot)$ is the distance under a chosen norm. Recent works (Lu & Song, 2024; Geng et al., 2025; Zheng et al., 2025) have further improved the method.

Recent works on real-time interactive video generation (Yin et al., 2025; Huang et al., 2025a) adopt a simplified variant of CD, which trains the student $G_\theta$ with direct regression: $\theta^* = \min_\theta \mathbb{E}_{t,x_t}[||G_\theta(x_t, t) - x_0||^2]$, where $x_t$ and $x_0$ lie on the same PF-ODE trajectory of the teacher model. We refer to this method as *ODE distillation* in the sequel.

### 2.4. Score Distillation

Score distillation (Wang et al., 2023; Luo et al., 2023b) distills a multi-step diffusion model into a few-step student model by matching the student's generative distribution $p_\theta(\tilde{x})$ to that of the data. A commonly used instantiation is Distribution Matching Distillation (DMD) (Yin et al., 2024),

which minimizes the KL divergence between the student and data distributions by descending along its gradient

$$\nabla_\theta \mathbb{E}_t[D_{\text{KL}}(p_{\theta,t} || p_{\text{data},t})]$$
$$= -\mathbb{E}_{\tilde{x},t,\tilde{x}_t}[(s_{\text{real}}(\tilde{x}_t, t) - s_{\text{fake}}(\tilde{x}_t, t))\frac{\partial \tilde{x}}{\partial \theta}], \quad (2)$$

where $\tilde{x} \sim p_\theta(\tilde{x})$ is generated by the student, and $\tilde{x}_t \sim q_{t|0}(\tilde{x}_t | \tilde{x})$ is a noised version of $\tilde{x}$ that induces the distribution $p_{\theta,t}(\tilde{x}_t)$. Herein, a frozen diffusion model $s_{\text{real}}$ is used to predict the score of $\tilde{x}_t$ under the noisy data distribution $p_{\text{data},t}(\tilde{x}_t)$, while an online-trainable diffusion model $s_{\text{fake}}$ predicts the score under $p_{\theta,t}(\tilde{x}_t)$.

## 3. Method

### 3.1. Limitations of Existing Methods

As described in Sec. 2.2, real-time interactive video generation requires a few-step autoregressive generator. The most widely used strategy, exemplified by CausVid (Yin et al., 2025) and Self Forcing (Huang et al., 2025a), adopts asymmetric distillation: given a pretrained bidirectional video diffusion model, one distills a few-step autoregressive student generator. Compared to standard step-distillation, beyond the shared *sampling-step gap*, i.e., reducing multi-step sampling to few-step sampling, a more fundamental challenge lies in the *architectural gap*: converting a bidirectional model with full attention (Peebles & Xie, 2023) into a causal attention architecture that conditions solely on past context, with no access to future frames.

Although the current state-of-the-art (SOTA) in autoregressive video diffusion distillation, Self Forcing (Huang et al., 2025a), achieves strong performance, it still falls short of standard DMD, which distills a few-step bidirectional student from a bidirectional video diffusion model. As shown in Fig. 1, Self Forcing is substantially worse than standard DMD (Yin et al., 2024) in terms of vision quality, dynamic degree, and instruction following. This gap suggests that existing autoregressive diffusion distillation pipelines remain suboptimal, motivating a further investigation of the underlying causes and more effective strategies.

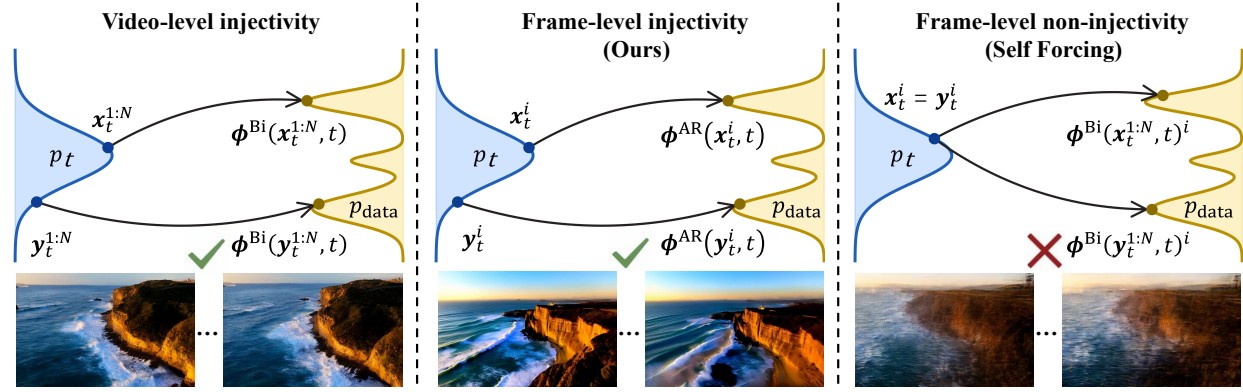

**(a)** Bidirectional teacher, bidirectional student     **(b)** AR teacher, AR student     **(c)** Bidirectional teacher, AR student

*Figure 3.* **Necessary principle for ODE initialization and why Self Forcing is flawed.** ODE distillation requires injective paired data. (a) Standard ODE distillation, which distills a *bidirectional* teacher to a *bidirectional* student, satisfies this requirement at the video level. (b) For an *AR* student, injectivity must hold at the frame level: each noisy frame maps to a unique clean frame via the PF-ODE of the *AR* teacher. (c) In contrast, Self Forcing distills an *AR* student from a *bidirectional* teacher, where the same noisy frame corresponds to multiple distinct clean frames, violating frame-level injectivity and results in blurred videos after ODE distillation. See Sec. 3.2 for details.

### 3.2. Analysis: Suboptimality of Existing Methods

In this section, we analyze the reason for the performance degradation observed in existing methods and identify the key principle required to address it.

**Recap of the Self Forcing Pipeline.** The current SOTA Self Forcing (Huang et al., 2025a) adopts a two-stage distillation strategy. Given a bidirectional diffusion model, it first applies an ODE distillation to bridge the architectural gap and enable few-step sampling. Specifically, the bidirectional model samples along its PF-ODE trajectory, and the target autoregressive model $G_\theta$ learns to map noisy intermediates to the clean video. The training objective is

$$\theta^* = \min_\theta \mathbb{E}_{t, \boldsymbol{x}_t^{1:N}, i}[||G_\theta(\boldsymbol{x}_t^i, \boldsymbol{x}_t^{<i}, t) - \boldsymbol{x}_0^i||^2], \quad (3)$$

where $i \sim \mathcal{U}(1, N)$, $(\boldsymbol{x}_t^{1:N}, \boldsymbol{x}_0^{1:N})$ lie on the same PF-ODE trajectory of the bidirectional teacher model, and $t \in \mathcal{S}$ denotes a predefined set of timesteps used for sampling. Building on this ODE distillation stage, it then further applies asymmetric DMD, using the resulting model to initialize the autoregressive student and the bidirectional base model as the teacher. In what follows, we examine how each stage contributes to closing the architectural gap and whether it is theoretically well aligned with this objective.

**DMD stage in Self Forcing does not address the architectural gap.** We first examine whether the DMD stage can bridge the architectural gap. We initialize the autoregressive student with a few-step bidirectional model distilled via standard DMD, which removes the sampling-step gap while retaining only the architectural gap. As shown in Fig. 2, despite eliminating the sampling-step gap, performance remains significantly worse than the standard DMD. This indicates that a large architectural gap at initialization

cannot be resolved by the subsequent DMD stage. Consequently, in the two-stage design of Self Forcing, *it is the ODE distillation stage that is expected to bridge the architectural gap.* We next analyze its theoretical soundness.

**Frame-level injectivity as a necessary principle for ODE initialization.** We begin by identifying the necessary condition that ODE distillation must satisfy. For regressive MSE-loss-based ODE distillation to be well-defined, the paired data must be injective (Liu et al., 2022), meaning that at any timestep, each noisy sample corresponds to a *unique* clean sample in the sample space. In the setting where a bidirectional student is distilled from a bidirectional teacher, this injectivity naturally holds by nature at the video level due to the injectivity of diffusion PF-ODE. Formally, for any noisy video $\boldsymbol{x}_t^{1:N}$, there exists a *unique* clean video $\boldsymbol{x}_0^{1:N}$ in the sample space, such that $\boldsymbol{x}_0^{1:N} = \boldsymbol{\phi}^{\mathrm{Bi}}(\boldsymbol{x}_t^{1:N}, t)$, where $\boldsymbol{\phi}^{\mathrm{Bi}} : (\boldsymbol{x}_t^{1:N}, t) \mapsto \boldsymbol{x}_0^{1:N}$ denotes the PF-ODE flow map of the bidirectional model (see Fig. 3a), and is exactly what the student model learns to fit. However, in autoregressive video models, frames are generated sequentially. This shifts the injectivity requirement from the entire video $\boldsymbol{x}_0^{1:N}$ to each individual frame $\boldsymbol{x}_0^i$, $i \in \{1, \ldots, N\}$. Specifically, for any noisy frame $\boldsymbol{x}_t^i$, there must exist a *unique* corresponding clean frame $\boldsymbol{x}_0^i$ in the sample space such that $\boldsymbol{x}_0^i = \boldsymbol{\phi}^{\mathrm{AR}}(\boldsymbol{x}_t^i, t)$, where $\boldsymbol{\phi}^{\mathrm{AR}} : (\boldsymbol{x}_t^i, t) \mapsto \boldsymbol{x}_0$ denotes PF-ODE flow map of the autoregressive diffusion model that the student model learns (see Fig. 3b). We formalize this requirement below:

**Definition 3.1** (Frame-level injectivity). For the mapping $\boldsymbol{\phi}^{\mathrm{AR}} : (\boldsymbol{x}_t^i, t) \mapsto \boldsymbol{x}_0^i$, frame-level injectivity holds if $\forall t \in (0, 1]$, for any two noisy videos $\{\boldsymbol{x}_t^j\}_{j=1}^N$, $\{\boldsymbol{y}_t^j\}_{j=1}^N$:

$$\forall i \in [N], \boldsymbol{x}_t^i = \boldsymbol{y}_t^i \Rightarrow \boldsymbol{\phi}^{\mathrm{AR}}(\boldsymbol{x}_t^i, t) = \boldsymbol{\phi}^{\mathrm{AR}}(\boldsymbol{y}_t^i, t), \quad (4)$$

i.e., $\phi^{\mathrm{AR}}(\boldsymbol{x}_t^i, t)$ is a well-defined function that maps $\boldsymbol{x}_t^i$ to the $i$-th clean frame (we omit the conditional history frames for brevity).

If the condition in Eq. (4) is violated, the regressive student trained via Eq. (3) cannot recover the teacher's flow map, but instead collapses to the conditional expectation (Bishop & Nasrabadi, 2006):

$$G_\theta^*(\boldsymbol{x}_t^i, \boldsymbol{x}_t^{<i}, t) = \mathbb{E}[\boldsymbol{x}_0 | \boldsymbol{x}_t^i, \boldsymbol{x}_t^{<i}, t]. \tag{5}$$

Intuitively, learning a conditional mean averages over frames, which manifests as blurred visual results.

**Current ODE initialization in Self Forcing violates frame-level injectivity.** Current ODE distillation in Self Forcing employs a *bidirectional* teacher to distill an *autoregressive* student. We show that this design violates the frame-level injectivity condition in Eq. (4), rendering the distillation fundamentally flawed.

As discussed above, the PF-ODE trajectory induced by a bidirectional model is injective only at the *video level*, but not at the *frame level*. We theoretically demonstrate that this leads to a non-negligible probability that the same noisy frame $\boldsymbol{x}_t^i$ corresponds to multiple distinct clean frames (see Fig. 3c and Lemma 3.2). Formally, for a fixed timestep $t$, there exist $\boldsymbol{x}_t^i = \boldsymbol{y}_t^i$ yet $\boldsymbol{x}_0^i \neq \boldsymbol{y}_0^i$ in the paired data sample space, and such collisions occur on a set of non-zero measure. Consequently, the frame-level injectivity condition in Eq. (4) is violated with non-negligible probability. This shows that Self Forcing's ODE distillation, which trains an autoregressive student from a bidirectional teacher, is theoretically misaligned. We formalize this issue in the following lemma:

**Lemma 3.2** (Frame-level non-injectivity of PF-ODE, informal). *Let $\boldsymbol{x}_t^{1:N}$ satisfy the PF-ODE in Eq. (1) of a bidirectional diffusion model. Denote $\boldsymbol{x}_t^i$ as its $i$-th frame, and let $\boldsymbol{x}_t^{\mathrm{other}} := \boldsymbol{x}_t^{[N]\setminus\{i\}}$ denote the remaining frames. Define the flow map $\phi^{\mathrm{Bi}} : (\boldsymbol{x}_t^{1:N}, t) \mapsto \boldsymbol{x}_0^{1:N}$. If $\phi^{\mathrm{Bi}}(\boldsymbol{x}_t^{1:N}, t)^i$ is not a.e. constant with respect to $\boldsymbol{x}_t^{\mathrm{other}}$, then*

$$\forall t \in (0, 1], \, \forall \boldsymbol{x}_t^{1:N} \in \mathbb{R}^d, \, \exists \boldsymbol{y}_t^{1:N} \in \mathbb{R}^d, such that$$
$$\boldsymbol{y}_t^i = \boldsymbol{x}_t^i, and \, \phi^{\mathrm{Bi}}(\boldsymbol{x}_t^{1:N}, t)^i \neq \phi^{\mathrm{Bi}}(\boldsymbol{y}_t^{1:N}, t)^i. \tag{6}$$

*Moreover, $\mathbb{P}\big(\mathrm{Var}\big(\phi^{\mathrm{Bi}}(\boldsymbol{x}_t^{1:N}, t)^i \mid \boldsymbol{x}_t^i, t\big) > 0\big) > 0$. For a rigorous formalization and proof, see Appendix B.1.*

This implies that $\phi^{\mathrm{Bi}}(\cdot, \cdot)^i$ is not a well-defined function. A key intuition for this issue is that a bidirectional diffusion model denoises the $i$-th frame using all frames. Thus, even with $\boldsymbol{x}_t^i$ fixed, different $\boldsymbol{x}_t^{>i}$ can yield different $\boldsymbol{x}_0^i$. In Self Forcing, the autoregressive student is supervised without $\boldsymbol{x}_t^{>i}$, causing information loss and thus violating Eq. (4).

Similar to the issue identified in Eq. (5), this frame-level non-injectivity prevents the autoregressive student model from recovering the teacher's true flow map:

**Proposition 3.3** (Distribution mismatch in current Self Forcing ODE distillation, proof in Appendix B.1). *Using the notation of Lemma 3.2, consider training a causal frame-wise model $G_\theta : (\boldsymbol{x}_t^i, t) \mapsto \boldsymbol{x}_0^i$ with the MSE regression target $\theta^* = \min_\theta \mathbb{E}_{\boldsymbol{x}_t^{1:N}, t}\left[\left\|G_\theta(\boldsymbol{x}_t^i, t) - \boldsymbol{x}_0^i\right\|^2\right]$ (we omit the conditional $\boldsymbol{x}_t^{<i}$ for brevity), where $(\boldsymbol{x}_t^{1:N}, \boldsymbol{x}_0^{1:N})$ is paired data from PF-ODE of a bidirectional diffusion model. Then the optimal solution does not follow the data distribution, i.e.,*

$$G_\theta^*(\boldsymbol{x}_t^i, t) = \mathbb{E}[\boldsymbol{x}_0^i \mid \boldsymbol{x}_t^i, t] \not\sim p_{\mathrm{data}}(\boldsymbol{x}_0^i). \tag{7}$$

As shown in Fig. 3c, this leads to blurry videos and is markedly inferior to standard ODE distillation with a bidirectional student in Fig. 3a.

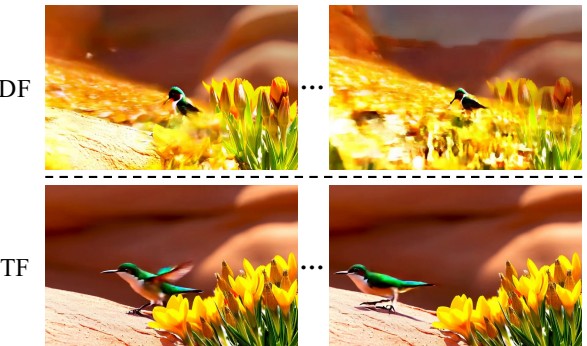

*Figure 4.* **TF vs. DF in AR diffusion training.** Contrary to common belief, DF leads to video collapse due to the training-inference gap, whereas TF produces higher visual quality.

### 3.3. Causal Forcing

Building on the above analysis, bridging the architectural gap requires ODE distillation to satisfy the frame-level injectivity condition in Eq. (4), which in turn requires an autoregressive diffusion model as the teacher. We therefore propose *Causal Forcing*, a three-stage method that sequentially consists of teacher forcing autoregressive diffusion training, causal ODE distillation, and asymmetric DMD.

**Stage 1: autoregressive diffusion training.** We begin by revisiting two standard training paradigms for AR diffusion models, namely teacher forcing (TF) and diffusion forcing (DF) (see Sec. 2.2), to obtain the AR diffusion model that serves as the teacher for subsequent ODE distillation. Somewhat surprisingly, and contrary to common belief, we find that *teacher forcing is more suitable than diffusion forcing for training AR diffusion models*, both theoretically and empirically. Specifically, when training the $i$-th frame $\boldsymbol{x}_t^i$, diffusion forcing is conditioned on heavily noised preceding frames $\boldsymbol{x}_t^{<i}$, whereas inference is conditioned on

clean preceding frames $\boldsymbol{x}_0^{<i}$. This discrepancy introduces a substantial training–inference distribution mismatch. We formalize this issue in the following proposition.

**Proposition 3.4** (Distribution mismatch in autoregressive diffusion forcing). *Under the notation of Sec. 2.2 and regularity conditions in Appendix B.2:*

$$\mathbb{E}_{\boldsymbol{y}\sim p_{data}(\boldsymbol{x}_0^{<i})}\Big[D_{\mathrm{KL}}\big(p_{DF}(\boldsymbol{x}_0^i \mid \boldsymbol{y}) \,\|\, p_{data}(\boldsymbol{x}_0^i \mid \boldsymbol{y})\big)\Big] > 0.$$

*That is, the model trained with autoregressive diffusion forcing does not follow the data distribution conditioned on the causal prefix $\boldsymbol{y}$. See Appendix B.2 for the proof.*

In contrast, teacher forcing conditions on clean preceding frames $\boldsymbol{x}_0^{<i}$ during training, thereby aligning the training objective with the inference process and eliminating this gap. The empirical comparisons in Fig. 4 further corroborate this theoretical analysis. For further discussion of diffusion forcing and recent alternatives (Wu et al., 2025; Po et al., 2025; Guo et al., 2025), see Appendix C.1. Accordingly, we adopt an autoregressive diffusion model trained via teacher forcing. ***Notably, this multi-step AR diffusion model already provides a substantially improved initialization for asymmetric DMD, but still exhibits abrupt artifacts. Please refer to Appendix C.2 for both qualitative and quantitative results.***

**Stage 2: causal ODE distillation.** With the above AR diffusion model as the teacher, we next perform causal ODE distillation. Firstly, we need to sample and store the PF-ODE trajectory $\{\boldsymbol{x}_t^i\}_{t\in\mathcal{S}\cup\{0\}}$ from the AR diffusion teacher at the required timesteps $\mathcal{S}$. To achieve this, we sample the clean frames $\boldsymbol{x}_{\mathrm{gt}}^{<i}$ from the real dataset and use the teacher to generate the current frame with ODE solver conditioned on these frames as history, starting from Gaussian noise $\boldsymbol{x}_T^i \sim \mathcal{N}(\boldsymbol{0}, \boldsymbol{I})$. Then, the student $G_\theta$ is trained to regress the clean target $\boldsymbol{x}_0^i$ from the intermediate noisy state $\boldsymbol{x}_t^i$, conditioned on the same history clean frames $\boldsymbol{x}_{\mathrm{gt}}^{<i}$:

$$\theta^* = \min_\theta \mathbb{E}_{\boldsymbol{x}_{\mathrm{gt}}^{<i},\, t\in\mathcal{S}, i, \boldsymbol{x}_t^i}[||G_\theta(\boldsymbol{x}_t^i, \boldsymbol{x}_{\mathrm{gt}}^{<i}, t) - \boldsymbol{x}_0^i||^2], \quad (8)$$

Notably, since the teacher here is autoregressive rather than bidirectional, its PF-ODE ensures the frame-level injectivity condition in Eq. (4) by nature. Therefore, our method avoids the collapse identified in Proposition 3.3 and enables the student to learn the flow map accurately. As shown in Fig. 3b, sampling such an AR teacher for ODE distillation indeed yields strong performance.

**Stage 3: asymmetric DMD.** Building upon the above causal ODE initialization for the AR student, we perform asymmetric DMD following Self Forcing. Exactly as in Self Forcing's DMD, this process lets the student model condition on its own generated prefix and perform self-rollout,

thereby reducing the training–inference gap. As shown in Fig. 5, DMD with our causal ODE initialization yields a final model that substantially outperforms Self Forcing, indicating that the architectural gap is effectively resolved.

### 3.4. Extension to Causal Consistency Models

Beyond the score-distillation paradigm mentioned above, since ODE distillation can be viewed as a simplified form of consistency distillation (CD), our perspective also naturally extends to CD. In this section, we present the first causal CD framework and further show that it outperforms the asymmetric CD that uses a bidirectional model as a teacher.

Specifically, we use the aforementioned native autoregressive diffusion model as the teacher, and train the causal consistency model $G_\theta$ via teacher forcing as follows:

$$\theta^* = \min_\theta \mathbb{E}_{\boldsymbol{x}_{\mathrm{gt}},\boldsymbol{\epsilon},t,i}[w(t)d(G_\theta(\boldsymbol{x}_t^i, \boldsymbol{x}_{\mathrm{gt}}^{<i}, t),$$
$$G_{\theta^-}(\hat{\boldsymbol{x}}_{t-\Delta t}^i, \boldsymbol{x}_{\mathrm{gt}}^{<i}, t - \Delta t))], \quad (9)$$

where $\hat{\boldsymbol{x}}_{t-\Delta t}^i$ is obtained by solving ODE from $\boldsymbol{x}_t^i$ using the autoregressive teacher model conditioned on clean prefix $\boldsymbol{x}_{\mathrm{gt}}^{<i}$, and other notations follow Sec. 2.3. Consistent with Sec. 3.3, this approach leverages the frame-level injectivity of the native AR teacher, enabling the student to learn the correct flow map. In contrast, asymmetric CD teacher violates this injectivity as stated in Lemma 3.2, leading to collapse. As shown in Fig. 10 in Appendix D, our causal CD substantially outperforms asymmetric CD.

Such causal CD may also serve as a substitute for causal ODE distillation, providing a strong initialization for the DMD stage. We leave this to future work.

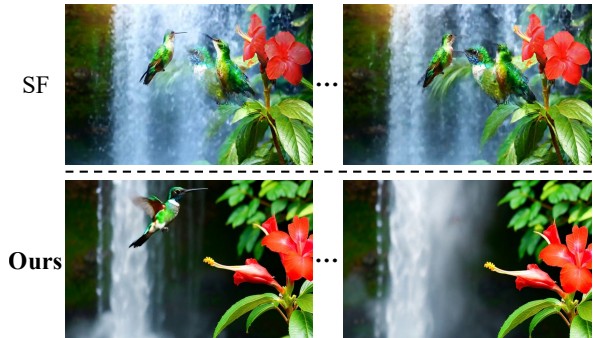

*Figure 5.* **Performance comparison between Self Forcing (SF) and ours.** DMD with Self Forcing's ODE initialization shows weaker dynamics and artifacts, whereas with causal ODE initialization, it achieves stronger dynamics with higher visual fidelity.

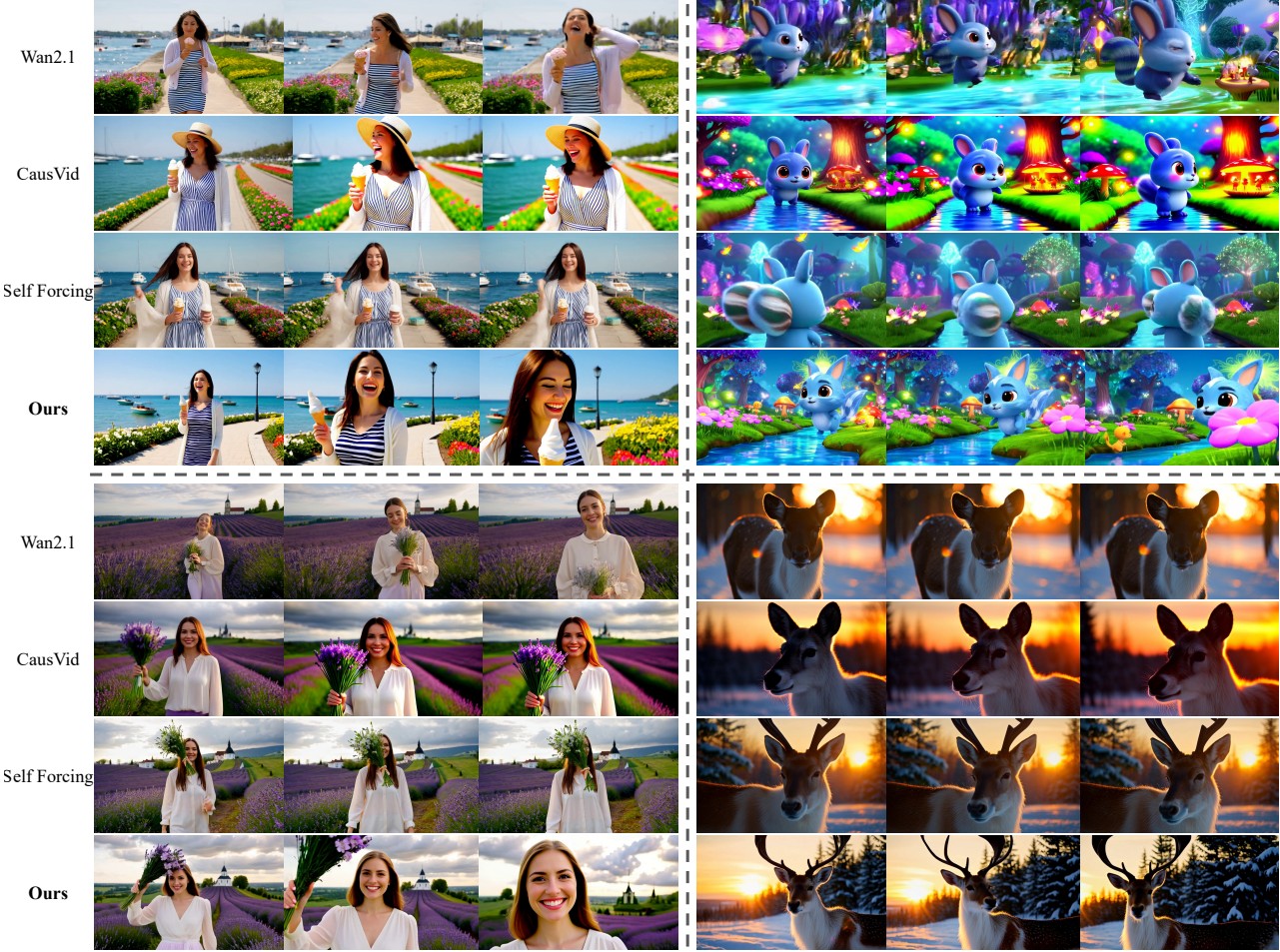

*Figure 6.* **Qualitative comparisons with existing methods.** Our method achieves substantially higher dynamics and better visual quality than existing distilled autoregressive video models (Causvid and Self Forcing), while matching or even surpassing bidirectional diffusion models (Wan2.1). *More video demos and all the prompts used in this paper are provided in the supplementary materials.*

## 4. Experiments

### 4.1. Setup

**Implementation details.** Following Self Forcing (Huang et al., 2025a), we adopt Wan2.1-T2V-1.3B (Wan et al., 2025) as our base model to fine-tune from, which generates 81-frame videos at a resolution of $832 \times 480$. We first train an autoregressive diffusion model with teacher forcing for 2K steps on a 3K dataset $\mathcal{D}_{Bi}$ synthesized by the base bidirectional model. When constructing $\mathcal{D}_{Bi}$, we also store noisy intermediates for the baseline ODE distillation for ablation. We then use the autoregressive diffusion model as the teacher to sample 3K causal ODE-trajectories $\mathcal{D}_{Causal}$ and perform causal ODE distillation for 1K steps. The resulting model initializes asymmetric DMD, trained on Vid-ProM (Wang & Yang, 2024) for 750 steps until convergence under the same protocol as Self Forcing. Following Self Forcing, we implement all methods in a chunk-wise manner, where each chunk contains 3 latent frames. For causal CD in the following ablation section, we adopt the LCM (Luo

et al., 2023a) setting. See more details in Appendix D.

**Evaluation.** We adopt VBench (Huang et al., 2024) as our primary evaluation benchmark following Self Forcing. For overall visual assessment, we employ VisionReward (Xu et al., 2024), which correlates well with human judgments, and we additionally report VisionReward's Instruction Following sub-score to measure instruction adherence. Notably, VisionReward scores can be negative, but higher values are always better. Since many VBench prompts involve minimal motion, we further curate a 100-prompt set with rich motion and complex actions, provided in the supplementary materials. We evaluate VisionReward, Instruction Following, and Dynamic Degree on this 100-prompt set. For readability, all these metrics are scaled by 100. Additionally, we conduct a user study with 10 participants on 10 prompts, where users rank the overall video quality across all methods. Finally, to assess real-time capability, we report throughput and latency on a single H100 GPU by FPS and seconds, respectively. See more details in Appendix D.

*Table 1.* **Quantitative comparisons with existing methods.** Our method consistently outperforms all baselines across all metrics. Dynamic., Vision., Instruct., and Rating denote Dynamic Degree, VisionReward, Instruction Following, and user rating, respectively.

| Model | Throughput ↑ | Latency ↓ | Total↑ | Quality↑ | Semantic↑ | Dynamic.↑ | Vision.↑ | Instruct.↑ | Rating↓ |
|---|---|---|---|---|---|---|---|---|---|
| *Bidirectional Video Diffusion Models* | | | | | | | | | |
| LTX-1.9B (HaCohen et al., 2024) | 8.98 | 13.5 | 79.83 | 81.88 | 71.62 | 46 | $-6.218$ | $-38$ | 6.40 |
| Wan2.1-1.3B (Wan et al., 2025) | 0.78 | 103 | 83.37 | 84.30 | 79.65 | 61 | 5.275 | 42 | 2.29 |
| *Autoregressive Video Diffusion Models* | | | | | | | | | |
| NOVA (Deng et al., 2024) | 0.88 | 4.1 | 80.31 | 80.66 | 78.92 | 46 | $-7.381$ | $-16$ | 8.41 |
| Pyramid Flow (Jin et al., 2024) | 6.70 | 2.5 | 80.75 | 83.41 | 70.11 | 16 | 4.055 | $-2$ | 6.11 |
| SkyReels-V2-1.3B (Chen et al., 2025) | 0.49 | 112 | 81.97 | 83.96 | 74.01 | 37 | 3.584 | 32 | 6.57 |
| MAGI-1-4.5B (Teng et al., 2025) | 0.19 | 282 | 78.88 | 81.67 | 67.72 | 42 | 0.773 | 8 | 6.44 |
| *Distilled Autoregressive Video Models* | | | | | | | | | |
| CausVid (Yin et al., 2025) | **17.0** | **0.69** | 81.33 | 83.98 | 70.72 | 62 | 5.741 | 12 | 4.27 |
| Self Forcing (Huang et al., 2025a) | **17.0** | **0.69** | 83.74 | 84.48 | 80.77 | 57 | 5.820 | 48 | 2.87 |
| Causal Forcing (**Ours**) | **17.0** | **0.69** | **84.04** | **84.59** | **81.84** | **68** | **6.326** | **56** | **1.64** |

## 4.2. Results

**Performance comparison with existing models.** We compare our model against baselines of comparable scale, including bidirectional video diffusion models Wan2.1-1.3B (Wan et al., 2025), LTX-1.9B (HaCohen et al., 2024), autoregressive video diffusion models NOVA (Deng et al., 2024), Pyramid Flow (Jin et al., 2024), SkyReels-V2-1.3B (Chen et al., 2025), MAGI-1-4.5B (Teng et al., 2025), and distilled autoregressive video models Causvid (Yin et al., 2025) and Self Forcing (Huang et al., 2025a).

As shown in Tab. 1, our method consistently outperforms all baselines across all metrics, achieving the best dynamic degree, visual quality, and instruction following ability. Compared to bidirectional diffusion models with a similar parameter scale, our method matches the performance of the SOTA Wan2.1 and even surpasses it, while delivering a 2079% higher throughput and substantially faster inference. In comparison with existing autoregressive diffusion models, our method improves over the best baseline by 47.8% in dynamic degree, 56.0% in VisionReward, and 75.0% in instruction following. Compared to distilled autoregressive video models, we maintain the same exceptionally high throughput and outperform the current SOTA Self Forcing by 19.3% in dynamic degree, 8.7% in VisionReward, and 16.7% in instruction following. Qualitative results in Fig. 6 align with the quantitative findings, showing that our method significantly surpasses the SOTA distilled autoregressive models. Notably, the baseline distilled autoregressive models both perform at least 3K steps of ODE initialization before DMD, the same as our method. Thus, our method uses exactly the same training budget, yet delivers substantial improvements.

**Ablation studies.** We compare different strategies under autoregressive diffusion training, score distillation, and consistency distillation (CD). For autoregressive diffusion training, Self Forcing's ODE initialization, and CD, we use

the $\mathcal{D}_{\text{Bi}}$ dataset; for our causal ODE initialization, we use $\mathcal{D}_{\text{Causal}}$. Since both datasets are internal synthetic data using the same prompts, we ensure that the data quality is identical, thus guaranteeing the fairness of the comparison. We report all results under the chunk-wise setting. In addition, for ODE initialization and DMD, we also report results under the frame-wise setting. See more details in Appendix D.

Tab. 2 shows that during autoregressive diffusion training, teacher forcing outperforms diffusion forcing across all metrics, with VisionReward improving by 111.2%, consistent with Fig. 4. Diffusion forcing attains a higher dynamic degree, but it largely stems from the collapse that pathologically inflates the motion metric. For ODE initialized DMD, our causal ODE initialization substantially outperforms Self Forcing's ODE initialization. Under the chunk-wise setting, DMD with our causal ODE initialization improves VisionReward by 90.0%, dynamic degree by 183.3%, and instruction following by 47.4%. This improvement is even more pronounced under the frame-wise setting, with a 3100% improvement in dynamic degree and a 218.0% increase in VisionReward. This is consistent with the qualitative results in Fig. 5, demonstrating that causal ODE distillation provides the correct initialization for DMD. We also compare our causal CD with the asymmetric CD, where causal CD improves VisionReward by 9.781 and instruction following by 60. Qualitative visualizations are provided in Appendix D Fig. 10. Notably, our current CD is still a rudimentary instantiation that directly adopts vanilla LCM (Luo et al., 2023a), and therefore underperforms score distillation. Nevertheless, our formulation paves the way for future work (Lu & Song, 2024; Zheng et al., 2025).

## 5. Conclusion

In this paper, we identify the limitations of existing methods for autoregressive video diffusion distillation and clarify that bridging the architectural gap is essential. Focusing on

*Table 2.* **Ablation study.** Tot., Qua., Sem., Dy., Vis., and Inst. denote Total, Quality, Semantic VBench score, Dynamic Degree, VisionReward, and Instruction Following, respectively.

| Method | Tot.↑ | Qua.↑ | Sem.↑ | Dy.↑ | Vis.↑ | Inst.↑ |
|---|---|---|---|---|---|---|
| *Autoregressive Diffusion Training* | | | | | | |
| Diffusion Forcing | 81.76 | 82.52 | 78.71 | **60** | 1.583 | 30 |
| Teacher Forcing | **82.12** | **82.73** | **79.67** | 50 | **3.343** | **32** |
| *Score Distillation (Chunk-wise)* | | | | | | |
| Self Forcing's ODE + DMD | 82.00 | 82.18 | 81.29 | 24 | 3.330 | 38 |
| Causal ODE + DMD | **84.04** | **84.59** | **81.84** | **68** | **6.326** | **56** |
| *Score Distillation (Frame-wise)* | | | | | | |
| Self Forcing's ODE + DMD | 81.83 | 82.66 | 78.50 | 2 | 1.951 | – 4 |
| Causal ODE + DMD | **83.75** | **84.35** | **81.37** | **64** | **6.204** | **42** |
| *Consistency Distillation* | | | | | | |
| Asymmetric CD | 79.07 | 79.99 | 75.37 | **59** | – 7.983 | – 42 |
| Causal CD | **81.48** | **82.13** | **78.88** | 51 | **1.798** | **18** |

the ODE initialization, we show that a fundamental requirement is frame-level injectivity that existing methods violate. Building on this theoretical analysis, we propose *Causal Forcing*: we first train an autoregressive diffusion model via teacher forcing, and then use it as the teacher for ODE distillation to initialize the subsequent DMD stage. Experiments show that our method consistently outperforms all baselines across all metrics, demonstrating its effectiveness.

# Acknowledgements

This work was supported by NSFC Projects (Nos. 62595773, 62522609, 62550004, 92470118, 92370124, U25B6003, 62350080), the Beijing Natural Science Foundation (No. L247030); partially supported by the Shandong Provincial Natural Science Foundation (No. ZR2022ZD01) and Beijing Natural Science Foundation L247011; supported by Tsinghua Institute for Guo Qiang, and the High Performance Computing Center, Tsinghua University. J. Zhu was also supported by the XPlorer Prize.

# Impact Statement

This paper presents work whose goal is to advance the field of Machine Learning. There are many potential societal consequences of our work, none of which we feel must be specifically highlighted here.

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

# A. Extended Related Work

**Video Generative Models.** Building on the tremendous success of diffusion models, many works have applied them to video generation (He et al., 2022; Ho et al., 2022; Singer et al., 2022; Blattmann et al., 2023a;b; Chen et al., 2023; Gupta et al., 2024; Zhao et al., 2024; Xing et al., 2024; Zhao et al., 2025b; 2022). With diffusion transformers (DiTs) demonstrating strong scalability (Bao et al., 2023; Peebles & Xie, 2023), many works have introduced DiT-based large-scale video models (Lin et al., 2024; Zheng et al., 2024; Polyak et al., 2024; Yang et al., 2024; HaCohen et al., 2024; Kong et al., 2024; Ma et al., 2025; Li et al., 2025), such as CogVideoX (Yang et al., 2024), Vidu (Bao et al., 2024) and Wan2.1 (Wan et al., 2025). Apart from the full-sequence diffusion models, some works adopt autoregressive next-token prediction to enable video generation (Wu et al., 2021; Hong et al., 2022; Wu et al., 2022; Weissenborn et al., 2019; Yan et al., 2021; Zhao et al., 2025c;a), such as NOVA (Deng et al., 2024) and VideoPoet (Kondratyuk et al., 2023). Video generation based on full-sequence diffusion models currently achieves better overall quality than autoregressive next-token prediction. However, full-sequence diffusion models must generate all frames in one shot, which incurs substantial latency and prevents displaying frames to users as they are produced, hindering interactivity and real-time use. In contrast, autoregressive models can generate videos in a streaming manner, enabling user interaction.

**Autoregressive Diffusion Models for Interactive Video Generation.** To combine the high quality of diffusion models with the interactivity of autoregressive models, recent works have proposed autoregressive diffusion video models (Jin et al., 2024; Deng et al., 2024; Teng et al., 2025; Chen et al., 2025). These models adopt a frame-wise autoregressive formulation while using diffusion within each frame, e.g., Pyramid Flow (Jin et al., 2024), MAGI-1 (Teng et al., 2025), and SkyReels-v2 (Chen et al., 2025). Such autoregressive diffusion models can display each frame to the user as soon as it is generated, and can adjust the conditioning for subsequent frames based on user feedback, enabling interactive generation. Nevertheless, interactivity typically requires real-time performance, meaning the generation speed should be comparable to the video playback rate. However, diffusion models rely on multi-step sampling and are therefore too slow to meet this requirement. To address this, recent works such as ASD (Yang et al., 2025b), CausVid (Yin et al., 2025) and Self Forcing (Huang et al., 2025a) introduce distillation strategies to obtain few-step generation models.

Such real-time, interactive video generation models are highly promising and have broad applications across many domains. One prominent application is video world modeling. HY-WorldPlay (Sun et al., 2025a), RELIC (Hong et al., 2025), Hunyuan-GameCraft-2 (Tang et al., 2025), and Yume-1.5 (Mao et al., 2025) train real-time interactive video models for realistic world simulation, allowing users to freely explore and take actions in the simulated environment. This interactive world-modeling paradigm further enables embodied intelligence, such as closed-loop control in Vidarc (Feng et al., 2025).

Another major application lies in entertainment and media, supporting interactive content generation (Sun et al., 2025b; Ki et al., 2026), including Knot Forcing (Xiao et al., 2025), Live avatar (Huang et al., 2025b), and Motionstream (Shin et al., 2025). Beyond interactivity, these autoregressive diffusion models have also been shown to excel at long-video generation, as demonstrated by Rolling Forcing (Liu et al., 2025), LongLive (Yang et al., 2025a), Self-Forcing++ (Cui et al., 2025), and Deep Forcing (Yi et al., 2025).

# B. Proofs of Propositions

We assume that all expectations appearing below are well-defined and finite, and that all probability density functions involved are integrable, which are mild regularity conditions standard in diffusion modeling (Song et al., 2020).

## B.1. The Flaw of Self Forcing's ODE Distillation

In this section, we first present a formal mathematical statement of Lemma 3.2 and provide its proof. Building on this result, we then prove Proposition 3.3.

**Lemma B.1** (Chunk-wise non-injectivity of PF-ODE). *Let $\boldsymbol{x}_t \in \mathbb{R}^d$ satisfy the PF-ODE $\mathrm{d}\boldsymbol{x}_t = \boldsymbol{v}_\theta(x_t, t)\,\mathrm{d}t$ with the unique solution. Define the flow map $\boldsymbol{\phi}: \mathbb{R}^d \times (0,1] \to \mathbb{R}^d$ by $\boldsymbol{\phi}(\boldsymbol{x}_t, t) = \boldsymbol{x}_0 \sim p_{data}(\boldsymbol{x}_0)$. Partition coordinates into $\boldsymbol{x}_t^u := (\boldsymbol{x}_t^{(m)}, \ldots, \boldsymbol{x}_t^{(n)})$ and $\boldsymbol{x}_t^z := \boldsymbol{x}_t^{[d] \setminus \{m,\ldots,n\}}$, with $k := n - m + 1 < d$. If $\boldsymbol{\phi}(\boldsymbol{x}_t, t)^u$ is not a.e. constant in $\boldsymbol{x}_t^z$ for each fixed $(\boldsymbol{x}_t^u, t)$, then*

$$\forall t \in (0,1], \ \forall \boldsymbol{x}_t \in \mathbb{R}^d, \ \exists \boldsymbol{y}_t \in \mathbb{R}^d, \text{such that } \boldsymbol{y}_t^u = \boldsymbol{x}_t^u, \text{ and } \boldsymbol{\phi}(\boldsymbol{y}_t, t)^u \neq \boldsymbol{\phi}(\boldsymbol{x}_t, t)^u. \tag{10}$$

*Moreover, if $\boldsymbol{x}_T \sim \mathcal{N}(\boldsymbol{0}, \boldsymbol{I})$ and $p_t(\boldsymbol{x}_t) > 0$ for Lebesgue-a.e. $\boldsymbol{x}_t$, then*

$$\mathbb{P}(\mathrm{Var}(\boldsymbol{\phi}(\boldsymbol{x}_t, t)^u \mid \boldsymbol{x}_t^u, t) > 0) > 0. \tag{11}$$

*Proof.* Fix $t \in (0, 1]$. Write any $\boldsymbol{x} \in \mathbb{R}^d$ as $\boldsymbol{x} = (\boldsymbol{x}^u, \boldsymbol{x}^z)$ with $\boldsymbol{x}^u \in \mathbb{R}^k$ and $\boldsymbol{x}^z \in \mathbb{R}^{d-k}$. Fix an arbitrary $\boldsymbol{x}_t \in \mathbb{R}^d$ and denote $\boldsymbol{u}_1 := \boldsymbol{x}_t^u$ and $\boldsymbol{z}_1 := \boldsymbol{x}_t^z$. Define the measurable map

$$\boldsymbol{f}_{\boldsymbol{u}_1, t} : \mathbb{R}^{d-k} \to \mathbb{R}^k, \qquad \boldsymbol{f}_{\boldsymbol{u}_1, t}(\boldsymbol{z}) := \boldsymbol{\phi}((\boldsymbol{u}_1, \boldsymbol{z}), t)^u. \tag{12}$$

By assumption, for this fixed $(\boldsymbol{u}_1, t)$, the function $\boldsymbol{z} \mapsto \boldsymbol{f}_{\boldsymbol{u}_1, t}(\boldsymbol{z})$ is *not* a.e. constant (with respect to Lebesgue measure on $\mathbb{R}^{d-k}$). We claim that then for every $\boldsymbol{z}_1 \in \mathbb{R}^{d-k}$ there exists some $\boldsymbol{z}_2 \in \mathbb{R}^{d-k}$ such that $\boldsymbol{f}_{\boldsymbol{u}_1, t}(\boldsymbol{z}_2) \neq \boldsymbol{f}_{\boldsymbol{u}_1, t}(\boldsymbol{z}_1)$. Indeed, if for some $\boldsymbol{z}_1$ one had $\boldsymbol{f}_{\boldsymbol{u}_1, t}(\boldsymbol{z}) = \boldsymbol{f}_{\boldsymbol{u}_1, t}(\boldsymbol{z}_1)$ for all $\boldsymbol{z}$, then $\boldsymbol{f}_{\boldsymbol{u}_1, t}$ would be everywhere constant, contradicting the assumption.

Choose such a $\boldsymbol{z}_2$ for the above $\boldsymbol{z}_1$, and set $\boldsymbol{y}_t := (\boldsymbol{u}_1, \boldsymbol{z}_2)$. Then $\boldsymbol{y}_t^u = \boldsymbol{x}_t^u$ while

$$\boldsymbol{\phi}(\boldsymbol{y}_t, t)^u = \boldsymbol{f}_{\boldsymbol{u}_1, t}(\boldsymbol{z}_2) \neq \boldsymbol{f}_{\boldsymbol{u}_1, t}(\boldsymbol{z}_1) = \boldsymbol{\phi}(\boldsymbol{x}_t, t)^u, \tag{13}$$

which proves the expression (10).

Combining the above result with the fact that $\boldsymbol{x}_T \sim \mathcal{N}(\boldsymbol{0}, \boldsymbol{I})$ and $\boldsymbol{x}_t$ admits a nondegenerate probability density, standard measure-theoretic arguments (Kallenberg, 1997; Evans, 2018) imply the following: for the above $\boldsymbol{z}_1, \boldsymbol{z}_2$, in a neighborhood of $\boldsymbol{z}_2$ there exist uncountably many $\boldsymbol{z}_k$, each of which maps to a distinct $\boldsymbol{\phi}(\boldsymbol{x}_t, t)^u$, just as $\boldsymbol{z}_2$ does. Equivalently, $\mathbb{P}(\mathrm{Var}(\boldsymbol{\phi}(\boldsymbol{x}_t, t)^u \mid \boldsymbol{x}_t^u, t) > 0) > 0$. □

We next prove Proposition 3.3. First, we formalize this in the following statement.

**Proposition B.2** (Distribution mismatch in chunk-wise regression). *Using the notation of Lemma B.1, and for each fixed $(\boldsymbol{x}_t^u, t)$, $\boldsymbol{\phi}(\boldsymbol{x}_t, t)^u$ is not a.e. constant in $\boldsymbol{x}_t^z$. Consider training a chunk-level model $G_\theta : \mathbb{R}^k \times (0, 1] \to \mathbb{R}^k$ with the regression target*

$$\theta^* = \min_\theta \mathbb{E}_{\boldsymbol{x}_t, t}\left[\|G_\theta(\boldsymbol{x}_t^u, t) - \boldsymbol{x}_0^u\|^2\right], \tag{14}$$

*where $\boldsymbol{x}_0 = \boldsymbol{\phi}(\boldsymbol{x}_t, t)$. Then the optimal solution is the conditional mean, which does not follow the chunk-wise data distribution, i.e.,*

$$G_\theta^*(\boldsymbol{x}_t^u, t) = \mathbb{E}[\boldsymbol{x}_0^u \mid \boldsymbol{x}_t^u, t] \nsim p_{\mathrm{data}}(\boldsymbol{x}_0^u). \tag{15}$$

Since DiT-based bidirectional diffusion models contain attention modules that are not constant as evidenced by attention-map experiments in prior works (Xi et al., 2025; Zhao et al., 2025d), the condition in Lemma B.1 holds: for each fixed $(\boldsymbol{x}_t^u, t)$, $\boldsymbol{\phi}(\boldsymbol{x}_t, t)^u$ is not a.e. constant in $\boldsymbol{x}_t^z$. We then prove Proposition B.2 using the expression (11).

*Proof.* Let $t \sim q(t)$ be the time sampling used in training and let $\boldsymbol{x}_0 = \boldsymbol{\phi}(\boldsymbol{x}_t, t)$. Denote

$$U := \boldsymbol{x}_t^u \in \mathbb{R}^k, \qquad Y := \boldsymbol{x}_0^u \in \mathbb{R}^k, \qquad \widehat{Y} := \mathbb{E}[Y \mid U, t]. \tag{16}$$

By the standard squared-loss regression result (Bishop & Nasrabadi, 2006), the minimizer satisfies

$$G_\theta^*(U, t) = \widehat{Y} = \mathbb{E}[\boldsymbol{x}_0^u \mid \boldsymbol{x}_t^u, t]. \tag{17}$$

It remains to show $\widehat{Y} \nsim Y$ (hence $\widehat{Y} \nsim p_{\mathrm{data}}(\boldsymbol{x}_0^u)$). Using the $L^2$-orthogonal projection identity (Bishop & Nasrabadi, 2006),

$$\mathbb{E}\|Y\|^2 = \mathbb{E}\|\widehat{Y}\|^2 + \mathbb{E}\|Y - \widehat{Y}\|^2 = \mathbb{E}\|\widehat{Y}\|^2 + \mathbb{E}[\mathrm{Var}(Y \mid U, t)], \tag{18}$$

where $\mathrm{Var}(Y \mid U, t) := \mathbb{E}[\|Y - \mathbb{E}[Y \mid U, t]\|^2 \mid U, t]$. By Lemma B.1 (expression 11), $\mathbb{P}(\mathrm{Var}(Y \mid U, t) > 0) > 0$, hence $\mathbb{E}[\mathrm{Var}(Y \mid U, t)] > 0$ and therefore

$$\mathbb{E}\|Y\|^2 > \mathbb{E}\|\widehat{Y}\|^2. \tag{19}$$

If $\widehat{Y}$ and $Y$ have the same distribution, then $\mathbb{E}\|\widehat{Y}\|^2 = \mathbb{E}\|Y\|^2$, a contradiction. Thus $\widehat{Y} \nsim Y$, i.e.,

$$G_\theta^*(\boldsymbol{x}_t^u, t) = \mathbb{E}[\boldsymbol{x}_0^u \mid \boldsymbol{x}_t^u, t] \nsim p_{\mathrm{data}}(\boldsymbol{x}_0^u). \tag{20}$$

□

## B.2. Distribution Mismatch in Autoregressive Diffusion Forcing

In this section, we first present the basic regularity assumptions and then prove Proposition 3.4.

Let $Y := \boldsymbol{x}_0^{<i}$, $X := \boldsymbol{x}_0^i$, and let $Z := \boldsymbol{x}_t^{<i}$ be obtained by independently noising each frame of $Y$ via the forward kernel $q_{t|0}(Z \mid Y)$ at some fixed $t > 0$. We have the following mild assumptions.

- **(A1)** The model yields the optimal conditional distribution under the diffusion training objective, denoted as $p_{\mathrm{DF}}(X \mid Z) =: p_{\mathrm{DF}}(\boldsymbol{x}_0^i \mid \boldsymbol{x}_t^{<i}) = p_{\mathrm{data}}(x \mid z)$. This is a trivial assumption widely used (Song et al., 2020).

- **(A2)** $X$ is not independent of $Y$ under $p_{\mathrm{data}}(X, Y)$, i.e., $p_{\mathrm{data}}(X \mid Y = y)$ is not $p_{\mathrm{data}}(Y)$-a.e. constant. The intuitive understanding of this assumption is that different frames within the same video are not independent, but are closely related.

- **(A3)** The posterior kernel $p_{\mathrm{data}}(Y \mid Z = y)$ is positive and non-degenerate on the support of $p_{\mathrm{data}}(Y)$. In particular, for $y \in \mathrm{supp}(p_{\mathrm{data}}(Y))$, the density of $p_{\mathrm{data}}(Y \mid Z = y)$ is positive.

*Proof.* By **(A1)**, querying the DF-trained model at a *clean* prefix value $y$ induces

$$p_{\mathrm{DF}}(x \mid y) = p_{\mathrm{DF}}(x \mid Z = y) = p_{\mathrm{data}}(x \mid Z = y). \tag{21}$$

Therefore, it suffices to prove

$$\mathbb{E}_Y \Big[ D_{\mathrm{KL}}\big( p_{\mathrm{data}}(X \mid Z = Y) \,\|\, p_{\mathrm{data}}(X \mid Y) \big) \Big] > 0. \tag{22}$$

We prove by contradiction. Assume for contradiction that the left-hand side of expression (22) equals $0$. This implies

$$p_{\mathrm{data}}(X \mid Z = y) = p_{\mathrm{data}}(X \mid Y = y) \qquad \text{for } p_{\mathrm{data}}(Y)\text{-a.e. } y. \tag{23}$$

Fix any measurable set $A$ in the sample space of $X$ and define

$$f_A(y) := \mathbb{P}_{\mathrm{data}}(X \in A \mid Y = y). \tag{24}$$

By Eq. (23), for $p_{\mathrm{data}}(Y)$-a.e. $y$,

$$\mathbb{P}_{\mathrm{data}}(X \in A \mid Z = y) = f_A(y). \tag{25}$$

On the other hand, since $Z$ is generated from $Y$ via independent noising, we have the Markov chain $X \to Y \to Z$ under $p_{\mathrm{data}}$. Thus, by the tower property (Kallenberg, 1997),

$$\mathbb{P}_{\mathrm{data}}(X \in A \mid Z = y) = \mathbb{E}_{\mathrm{data}}[\mathbb{P}_{\mathrm{data}}(X \in A \mid Y) \mid Z = y] \tag{26}$$

$$= \mathbb{E}_{\mathrm{data}}[f_A(Y) \mid Z = y]. \tag{27}$$

Combining Eq. (25) and Eq. (27) yields

$$f_A(y) = \mathbb{E}_{\mathrm{data}}[f_A(Y) \mid Z = y] \qquad \text{for } p_{\mathrm{data}}(Y)\text{-a.e. } y. \tag{28}$$

By the regularity conditions **(A3)**, the conditional expectation operator $Tf(y) := \mathbb{E}[f(Y) \mid Z = y]$ admits only $p_{\mathrm{data}}(Y)$-a.e. constant bounded fixed points. Applying this fact to Eq. (28) implies that $f_A$ is $p_{\mathrm{data}}(Y)$-a.e. constant for every measurable $A$. Hence $p_{\mathrm{data}}(X \mid Y = y)$ is $p_{\mathrm{data}}(Y)$-a.e. constant, i.e., $X \perp\!\!\!\perp Y$, which contradicts **(A2)**. Therefore, the contradiction assumption is false, and the expression (22) holds. Consequently,

$$\mathbb{E}_{y \sim p_{\mathrm{data}}(Y)} \Big[ D_{\mathrm{KL}}\big( p_{\mathrm{DF}}(X \mid y) \,\|\, p_{\mathrm{data}}(X \mid y) \big) \Big] > 0. \tag{29}$$

$\square$

*Table 3.* **Quantitative comparison of Autoregressive diffusion training strategies.** The recent works perform on par with teacher forcing in our setting.

| Method | Dynamic Degree↑ | VisionReward↑ | Instruction Following↑ |
|---|---|---|---|
| Teacher Forcing | 50 | **3.343** | **32** |
| PFVG (Wu et al., 2025) | **61** | 2.857 | **32** |
| BAgger (Po et al., 2025) | 53 | 2.715 | 28 |
| Resampling Forcing (Guo et al., 2025) | 51 | 3.336 | 22 |

## C. More Discussion of Our Method

### C.1. Further Remarks on Autoregressive Diffusion Training Strategies

In this section, we first provide further remarks on diffusion forcing, and then report results for other training strategies, including PFVG (Wu et al., 2025), BAgger (Po et al., 2025), and Resampling Forcing (Guo et al., 2025).

As stated in Proposition 3.4, applying diffusion forcing to autoregressive diffusion training is suboptimal. However, this does not render diffusion forcing useless. Specifically, diffusion forcing was originally introduced to train a bidirectional diffusion model for video continuation to enable long-video generation (Song et al., 2025). In this setting, continuation at inference time concatenates a clean prefix (the tail frames of the given video) with noise, which matches the training setup and thus avoids a train–inference mismatch. Therefore, this bidirectional diffusion forcing regime is not covered by our suboptimality claim. Moreover, diffusion forcing allows different frames to have different noise levels. Even in autoregressive diffusion training, this is not an issue, since each frame is actually trained independently. *The only practice we refute is conditioning on a noisy prefix for autoregressive diffusion training, as proposed by CausVid (Yin et al., 2025) and recent works (e.g., LiveAvatar (Huang et al., 2025b)).*

Apart from diffusion forcing and teacher forcing, we also experiment with several recent alternatives, including PFVG (Wu et al., 2025), BAgger (Po et al., 2025), and Resampling Forcing (Guo et al., 2025). However, as shown in Tab. 3, these methods provide no significant improvement over teacher forcing. Notably, most of them are primarily designed for long-video training and generation, so the limited gains in our 5s setting are understandable. We leave a deeper investigation of these strategies for future work.

### C.2. Multi-Step Autoregressive Diffusion as Initialization for Asymmetric DMD

In this section, we investigate directly using a teacher forcing-trained multi-step autoregressive diffusion model to initialize asymmetric DMD. We find that, compared to Self Forcing's ODE distillation initialization, multi-step autoregressive diffusion initialization yields substantial improvements in both dynamics and visual quality, as illustrated in Fig. 8 (middle vs. left).

Despite these improvements, the multi-step autoregressive diffusion model only narrows the bidirectional-to-causal architectural gap under multi-step sampling (e.g., 50 steps) and does not fully resolve it in the few-step regime. Specifically, under few-step sampling, autoregressive generation induces an additional mismatch in the conditional context: the $i$-th frame conditions on preceding frames $0 \sim i - 1$, whose quality degrades at low step counts, whereas training conditions on a high-quality ground-truth prefix. As a result, this degraded conditioning accumulates throughout autoregressive generation, causing error propagation across chunks. As illustrated in Fig. 7 (top), before the DMD stage, we evaluate the autoregressive diffusion model on 4-step generation only; it exhibits abrupt transitions between chunks, indicating that a substantial architectural gap remains in the few-step setting.

This analysis suggests that it's necessary to convert the multi-step autoregressive diffusion model to a few-step model before using it to initialize DMD. As illustrated in Fig. 7 (bottom), the causal ODE-distilled model exhibits stronger temporal consistency under few-step sampling, making it a more suitable DMD initialization. Consistently, Fig. 8 (middle vs. right) further shows that replacing multi-step autoregressive diffusion initialization with causal ODE initialization yields clear gains in the subsequent DMD stage. This is also supported by the quantitative results in Tab. 4, where causal ODE attains markedly better VisionReward and Instruction Following scores.

*Table 4.* **Quantitative comparison of DMD with different initializations.** DMD with Self Forcing's ODE initialization shows weak dynamics and low visual quality. Initializing with a Teacher Forcing-trained autoregressive diffusion model yields a large improvement, while causal ODE initialization achieves the best overall quality and VisionReward score.

| Method | Total↑ | Quality↑ | Semantic↑ | Dynamic Degree↑ | VisionReward↑ | Instruction Following↑ |
|---|---|---|---|---|---|---|
| Self Forcing's ODE + DMD | 82.00 | 82.18 | 81.29 | 24 | 3.330 | 38 |
| Autoregressive diffusion + DMD | 84.02 | **84.72** | 81.23 | 66 | 5.863 | 48 |
| Causal ODE + DMD | **84.04** | 84.59 | **81.84** | **68** | **6.326** | **56** |

*Figure 7.* **Performance comparison with 4-step generation before the DMD stage.** Without having reached the DMD stage yet, we directly compare the 4-step generation of the autoregressive diffusion model with the 4-step generation of the causal ODE-distilled model. Autoregressive diffusion exhibits inter-frame abrupt changes, indicating suboptimal causality under 4 steps, whereas the causal ODE–distilled model remains more stable.

*Figure 8.* **Performance comparison of DMD with different initialization.** DMD with Self Forcing's ODE initialization shows weak dynamics and abrupt artifacts. Initializing with TF-trained autoregressive diffusion brings a large improvement but still exhibits abrupt changes (e.g., two red flowers turning into one), whereas causal ODE initialization yields the highest quality and the most stable results.

| Self Forcing's ODE distillation, bidirectional initial model | Causal ODE distillation, causal initial model | Causal ODE distillation, bidirectional initial model |
|---|---|---|

*Figure 9.* **Student initialization is not the bottleneck of ODE distillation.** With causal ODE distillation, the student with a bidirectional initial model achieves similar performance to that with a causal initial model, both better than Self Forcing's ODE distillation.

### C.3. Causal ODE Distillation from Bidirectional Initial Model

Recall from Sec. 3.2 that we claim that we should adopt causal ODE distillation rather than Self Forcing's ODE distillation. For causal ODE distillation, the paired data $(\boldsymbol{x}_t^i, \boldsymbol{x}_0^i)$ are generated by an autoregressive diffusion model, and the student is initialized from this autoregressive diffusion model as well. For Self Forcing's ODE distillation, the paired data are generated by a bidirectional diffusion model, and the student is initialized from this bidirectional diffusion model as well. Though our main argument attributes the performance gap of these two methods to how the paired data are constructed, yet these two methods also differ in their initialization, raising a natural question: *is the observed difference in Fig. 3 mainly driven by the paired data construction, or by the initialization difference?*

To answer this question, we use paired data synthesized by the autoregressive diffusion model, i.e., $\mathcal{D}_{\text{Causal}}$ in Sec. 4, while initializing the student from the bidirectional diffusion model. As shown in Fig. 9, the resulting quality is comparable to initializing from the autoregressive diffusion model, still much better than that of the Self Forcing's ODE distillation. This indicates that the performance gap in ODE distillation is not primarily due to student initialization, but rather to the distillation setup: the teacher should be an autoregressive diffusion model rather than a bidirectional one.

## D. More Implementation Details

In this section, we provide more details of Sec. 4.

**Training details of our method.** We first construct a dataset $\mathcal{D}_{\text{Bi}}$ consisting of about 3K samples generated by Wan (bidirectional) (Wan et al., 2025) with the VidProM (Wang & Yang, 2024) prompts and train a teacher forcing autoregressive diffusion model for 2K steps. Next, we sample ODE trajectories from the autoregressive diffusion model to construct a causal ODE dataset $\mathcal{D}_{\text{Causal}}$ with 3K samples. Notably, since the causal ODE distillation conducts teacher forcing conditioned on the ground-truth clean data, we save the corresponding relationship between each data point in $\mathcal{D}_{\text{Causal}}$ and $\mathcal{D}_{\text{Bi}}$. Throughout training, including the teacher forcing autoregressive diffusion model and both ODE-initialization variants, we use either $\mathcal{D}_{\text{Bi}}$ or $\mathcal{D}_{\text{Causal}}$, both internally synthesized by the model, and using about the same prompts, thus guaranteeing that there is no gap in terms of data quality. This ensures the fairness of the comparison in the ablation study. Then, we perform causal ODE distillation on $\mathcal{D}_{\text{Causal}}$ for 1K steps via teacher forcing, conditioned on the corresponding clean data in $\mathcal{D}_{\text{Bi}}$. In this stage, the ODE student is initialized from the autoregressive diffusion teacher. Finally, we use this model as initialization for the standard asymmetric DMD, where $s_{\text{real}}$ is Wan2.1-14B, and $s_{\text{fake}}$ is Wan2.1-1.3B, strictly following the setting of Self Forcing (Huang et al., 2025a) to guarantee the fair comparison. For both the chunk-wise and frame-wise settings, we adopt the same overall formulation and pipeline.

Throughout all training stages, we use a batch size of 64 and the Adam optimizer with a learning rate of $2 \times 10^{-6}$, $\beta_1 = 0$, and $\beta_2 = 0.999$, while keeping all other settings identical to Self Forcing. During inference, we use 4-step sampling with timesteps shared by causal ODE initialization and asymmetric DMD, i.e., 1, 0.9375, 0.8333, and 0.625.

For the extended causal consistency distillation, we adopt the LCM (Luo et al., 2023a) scheme with 48 discretized timesteps, using the UniPC ODE solver with an EMA rate of 0.99. We train the model for 3K steps on $\mathcal{D}_{\text{Bi}}$, and inference also uses 4 steps, with the same timesteps as DMD. Notably, discrete-time consistency distillation requires the boundary condition $G_\theta(\boldsymbol{x}^i, \boldsymbol{x}_{\text{gt}}^{<i}, 0) \equiv \boldsymbol{x}^i$, which is typically implemented by introducing a wraped network $F_\theta$ to parameterize $G_\theta$:

$$G_\theta(\boldsymbol{x}^i, \boldsymbol{x}_{\text{gt}}^{<i}, t) = c_{\text{skip}}(t)\boldsymbol{x}^i + c_{\text{out}}(t)F_\theta(\boldsymbol{x}^i, \boldsymbol{x}_{\text{gt}}^{<i}, t), \tag{30}$$

where $c_{\text{skip}}(0) = 1$, $c_{\text{out}}(0) = 0$. However, since we use flow matching, i.e., a $v$-prediction parameterization for the diffusion

| Type | Generated video |
|------|-----------------|
| **Asymmetric CD** | 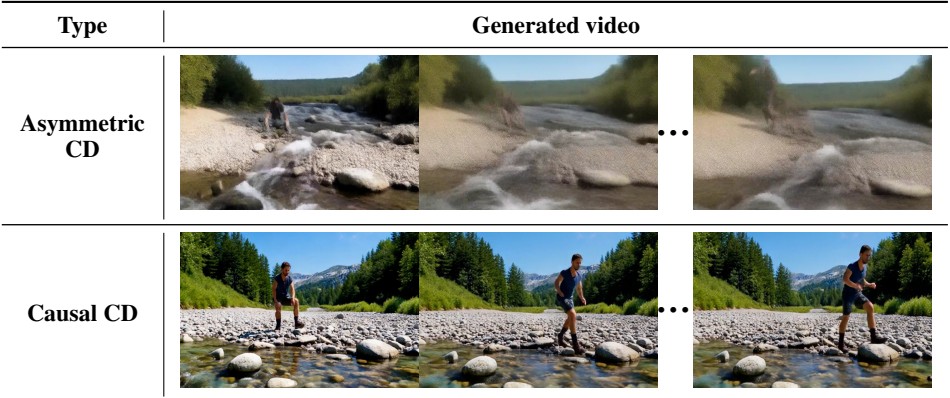 |
| **Causal CD** | |

*Figure 10.* **Comparison between asymmetric CD and causal CD.** Asymmetric CD appears highly blurry and exhibits abrupt artifacts, whereas causal CD results remain much better quality and more stable.

model $\boldsymbol{v}_\theta$, an $x_0$-prediction form for $G_\theta$ already satisfies the required boundary conditions, without any additional design:

$$G_\theta(\boldsymbol{x}^i, \boldsymbol{x}_{\text{gt}}^{<i}, t) = \boldsymbol{x}^i - t\boldsymbol{v}_\theta(\boldsymbol{x}^i, \boldsymbol{x}_{\text{gt}}^{<i}, t). \tag{31}$$

This simplified design may not be optimal and leaves substantial design space for further exploration (Geng et al., 2024; Lu & Song, 2024; Zheng et al., 2025), which we leave to future work. Fig. 10 visualizes that causal CD outperforms asymmetric CD, where asymmetric CD appears highly blurry and exhibits abrupt artifacts, whereas our results achieve better quality and are more stable. This agrees with Tab. 2 and highlights the necessity of a native causal teacher for autoregressive CD.

**Evaluation details.** In this section, we focus on the setups of Dynamic Degree, VisionReward, and Instruction Following. We use 100 prompts with rich action sequences and dynamics, provided in the supplementary material.

For Dynamic Degree, we use the official VBench evaluation code in the custom-input mode. Note that the Dynamic Degree reported in our table is evaluated on the 100-prompt motion set; however, when computing VBench's Total, Quality, and Semantic scores, the Dynamic Degree term is still evaluated on the standard VBench official prompts rather than inherited from our custom set. In addition, we use VisionReward to evaluate overall visual quality. Each sub-score can be positive or negative and lies in $[-1, 1]$, where $-1$ indicates the worst quality and $1$ the best. The final VisionReward score is computed as a weighted sum using the official weights. We additionally use VisionReward's prompt-alignment sub-score to evaluate instruction following, by querying VisionReward with the official prompt: *Does the video meet some of the requirements stated in the text "[[prompt]]"?*

**Performance comparison details.** All baselines use the same spatial resolution as Self Forcing. Throughput and latency on the H100 GPU for the baselines are taken directly from the Self Forcing paper.

**Ablation details.** For autoregressive diffusion training, both teacher forcing and diffusion forcing are trained for 3K steps. For Self Forcing's ODE initialization, we start from the bidirectional diffusion model and train for 3K steps. For causal ODE initialization, we first train a teacher forcing autoregressive diffusion model for 2K steps, and then run an additional 1K-step causal ODE distillation initialized from it. This aligns the training overall computation (both 3K in total) of the two ODE-initialization variants and ensures a fair comparison. Both CD variants are trained for 3K steps as well, each distilled using the teacher forcing autoregressive diffusion model as the teacher, ensuring a fair within-CD comparison. All other settings follow the main experiments.

