# OpenReview forum: "Causal Forcing: Autoregressive Diffusion Distillation Done Right for High-Quality Real-Time Interactive Video Generation"
_ICML.cc/2026/Conference — ICML 2026 regular_

### Official Review · Reviewer_8kgS · 2026-03-05

**Soundness:** 3
**Presentation:** 3
**Significance:** 3
**Originality:** 3
**Overall Recommendation:** 5
**Confidence:** 4

**Summary:**

The authors theoretically prove that existing methods, which distill an AR student from a bidirectional teacher, violate this frame-level injectivity. So they propose Causal Forcing, which employs teacher forcing to train an AR teacher for ODE initialization.

**Compliance With Llm Reviewing Policy:**

Affirmed.

**Final Justification:**

I thank the authors for their detailed and constructive rebuttal, which has helped clarify several of my initial concerns.
The rebuttal successfully addressed my main concerns, especially regarding theoretical consistency and experimental completeness. It has positively changed my evaluation by resolving misunderstandings and reinforcing confidence in both the theoretical and empirical contributions.
Overall, after considering both the paper and the rebuttal, I believe the work is sound, original, and practically significant. I support acceptance.

**Key Questions For Authors:**

See weekness.

**Limitations:**

Yes.

**Strengths And Weaknesses:**

Strengths
1. The workflow of the proposed method is presented clearly and logically, making the overall pipeline easy to follow from motivation to final distillation stage.
2. The motivation section effectively highlights the practical limitations of existing approaches, providing strong justification for the new framework.
3. The proposed method demonstrates superior empirical performance across multiple metrics and datasets, showing clear improvements over strong baselines in both quantitative scores and qualitative visual quality.

Weaknesses and Questions
1. In Lemma B.1, the provement starts from orthogonal coordinate system. But in Lemma 3.2, there is a strong dependency between frames. Will this influence?
Lemma B.1 derives its results under the assumption of an orthogonal coordinate system. However, Lemma 3.2 explicitly introduces strong inter-frame dependencies. Does the frame dependency break the orthogonality assumption in Lemma B.1?
If the orthogonality no longer holds exactly, it may affect the generality or tightness of the theoretical guarantees. The authors should provide a more general proof that relaxes the orthogonality assumption. Without this clarification, the theoretical foundation appears partially disconnected.
2. I remain doubtful whether the architectural gap between the teacher and student models is truly eliminated during the teacher-forcing (TF) stage.The underlying architectural differences may still persist in the internal representations. Consequently, this residual gap could be propagated to the student during the subsequent DMD stage, ultimately limiting the final performance ceiling.
3. The ablation study only reports quantitative metrics, which are insufficient to illustrate how the architecture gap manifests and how the proposed method mitigates it.
4. Can you provide more details about your dataset? What is the source of the data? Which prompt do you use to generate the 3K dataset?

---

> ### Author Rebuttal · Authors · 2026-03-31
>
> ## Author Response to Reviewer 8kgS
> We thank Reviewer 8kgS for the comments, but we would like to clarify that there are several **factual misunderstandings** here. The questions are addressed as follows.
>
> ### Q1: Is there a proof mismatch between Lemma B.1 and Lemma 3.2?
>
> We clarify that this is a **factual misunderstanding**. **Lemma B.1 does not assume orthogonality.** Instead, Lemma B.1 is based on inter-frame dependency, which is fully consistent with Lemma 3.2. Therefore, the two are fully aligned, with no mismatch or difference in assumptions. The detailed clarification is as follows:
> - Lemma B.1 does not assume orthogonality. We suspect the misunderstanding arises from the phrase "partition coordinates into $x_t^u$ and $x_t^z$" (Lines 709–710). In fact, it only means that the components are grouped into two index sets for notation. **This is purely a notational partition, not an orthogonal decomposition.**
> - Lemma B.1 uses the inter-frame dependency consistent with Lemma 3.2. It says that, "$\phi(x_t,t)^u$ is not a.e. constant in $x_t^z$ for each fixed $(x_t^u,t)$" (Lines 710-711). This means the recovered clean part $u$ depends on the other part $z$. Lemma 3.2 is exactly the frame version of this: $x_t^u$ is the $i$-th frame, and $x_t^z$ is the remaining frames. **So dependence on $x_t^z$ in Lemma B.1 is exactly the inter-frame dependency in Lemma 3.2.**
>
> In short, **the proof of Lemma B.1 uses dependency, not orthogonality**. Lemma B.1 is a mathematical formalization of the "frame" concept in Lemma 3.2, without any essential difference. We will revise the wording to make this clearer.
>
>
>
> ### Q2: Whether the architectural gap is eliminated
> We thank the reviewer for the comment. Compared with the incorrect initialization, our method yields a substantial improvement and even **matches or surpasses the bidirectional base model**. We therefore infer that our method has effectively bridged the architectural gap. Even if a small residual gap remains as suggested by the reviewer, the above empirical results indicate that this negligible residual gap does not adversely affect DMD.
>
>
>
>
> ### Q3: Whether the ablation study only reports quantitative metrics
> We clarify that the **qualitative visualizations for all ablation studies are already included in the original paper**. Specifically:
> - Lines 421–422 (left column) state that Fig. 4 presents the qualitative visualization of Diffusion Forcing versus Teacher Forcing;
> - Lines 432–433 (left column) state that Fig. 5 presents the qualitative visualization of Self Forcing vs. Causal Forcing;
> - Lines 438–439 (left column) state that Fig. 10 presents the qualitative visualization of Asymmetric CD vs. Causal CD.
>
> We thank the reviewer for the comment, and we will add the above references to the ablation tables to make them clearer.
>
>
> ### Q4: More details about the dataset
> We provided the detailed data information in Appendix D (Lines 965–966) of the original paper. Specifically, the dataset is generated by Wan 2.1 1.3B on 3K texts randomly sampled from VidProM. Thank you for the comment, and we will add this information into the main paper in the revised version.

---

> > ### Author Rebuttal · Reviewer_8kgS · 2026-04-02
> >
> > Thank you for your detailed and thoughtful response.
> >
> > Your clarifications have addressed my concerns, particularly regarding the relationship between Lemma B.1 and Lemma 3.2.
> > Overall, I am satisfied with the clarifications provided and will increase my score to accept.

---

> > > ### Author Response · Authors · 2026-04-02
> > >
> > > Thank you very much for the recognition of our work, for your positive feedback on our response, and for the update on the score. We sincerely appreciate your time and consideration.

---

### Official Review · Reviewer_gWzA · 2026-03-06

**Soundness:** 3
**Presentation:** 3
**Significance:** 4
**Originality:** 4
**Overall Recommendation:** 5
**Confidence:** 4

**Summary:**

The paper proposes an autoregressive diffusion distillation method for video generation, namely Causal Forcing. The work was motivated by addressing the architectural gap due to the pretrained bidirectional video diffusion models distilled into few-step autoregressive (AR) models. The contribution of this work is the proposed "frame-level injectivity", where each noisy frame must map to a unique clean frame under the PF-ODE of the AR teacher, as illustrated in Figure 3. The paper offers theoretical analysis in this work to show why this frame-level injectivity must hold. The experiments were conducted to compare against the conventional diffusion forcing, teacher forcing, and self-forcing.

**Compliance With Llm Reviewing Policy:**

Affirmed.

**Final Justification:**

The paper proposes an interesting method to handle the long-term video generation problem. The contribution of this work is the proposed causal forcing. The technical method looks solid, and the experimental results are also convincing. The rebuttal also clearly addresses the remaining concerns. Therefore, I am happy to recommend the paper to be accepted.

**Key Questions For Authors:**

How is the proposed Causal Forcing handling the long-term video generation, especially in relaying the problem to the frame-level
injectivity.

What is the limitation of the proposed Causal Forcing if the frame-level injectivity has already been addressed and the architectural gap is bridged? In this case, what is the bottleneck for the future video generation methods?

What are the insights on how to train a good Causal Forcing method, from the implementation perspective?

**Limitations:**

The paper talks about some of the limitations in the appendix, but not in a very clear way. One particular thing is how to use the proposed method to address this long-term video generation, which is the research direction of the video generation field. The paper also does not provide many video demonstration results. This makes the evaluations narrow.

**Strengths And Weaknesses:**

The primary contribution of this work is the proposed "frame-level injectivity", where each noisy frame must map to a unique clean frame under the PF-ODE of the AR teacher. The assumption makes sense, and the paper provides a theoretical analysis to show why this frame-level injectivity must hold for the auto-regressive models to work well. Thus, the motivation of this work is clear, and the paper offers a clear pathway to derive the solutions to adhere to the frame-level injectivity by focusing on correcting the ODE initialization. The paper is technically sound.

The paper is also clearly written and well-structured. Particularly, Figure 3 offers a clear illustration on what is the architectural gap and how that was related to the ODE initialization. The addressed problem here is significant as it looks into the training-test distribution difference that typically exists in the AR distillation methods.  The paper offers a very good insight into why this problem occurs and how to solve it.

The originality of this work is apparent, as it proposes this "frame-level injectivity", which is sort of neglected by the previous work. In the end, the implementation of Causal ODE distillation looks much simpler.

While the paper has the above strength, there are also several concerns that need to be addressed.

1. When talking about how the Frame-level injectivity should be enforced, the paper could start from the clear definition of ODE Distillation, along with the given equations. Then, this could better help understand why this existing ODE initialization in Self Forcing violates
frame-level injectivity.

2. The paper offers a strong theoretical analysis on motivating the frame-level injectivity, which is appreciated. In the end, I believe the implementation of this frame-level injectivity looks much simpler as shown in the Causal ODE distillation. This is definitely good. But it will be also nice for the paper to clearly mention this on how the problem was simplified. This part is something missing from the current work.

3. The paper shows the performance comparison between Self Forcing (SF) and ours in Figure 5. However, the background of the "ours" looks much blurrier compared to the Self Forcing (SF), though the foreground objects apparently improve a lot. Could the paper explain why this is the case and what the reason is for that?

4. The included video demo from the supplementary file looks good. However, it only compares against the CausVID and Self-forcing. How about other compared methods in this work? They should be included for video comparisons as well. Further, the paper could also provide the video demos for the ablation study comparison. All these demo results will truly help to check on the synthesized video quality. One particular thing I notice that the video generated by the CausVID is rather static, why the proposed method shows more dynamic generations. Could the paper also explain why the proposed method can better produce such dynamic video generations?

5. One more thing related to the experimental evaluations. The paper should also mention how the proposed method can handle the long-term video generation, which is apparently very important. The paper offers some discussions on this direction. But it does not show the experimental results for that, which could be enhanced. And it will also be very nice to offer more insights on whether the proposed work could address this long-term video generation. If not, what is the limitation. The discussion part in appendix is not sufficient at the moment.

Overall, this is a solid paper, with a blend of both theory and practice. However, the paper also needs to better address the raised questions and concerns.

---

> ### Author Rebuttal · Authors · 2026-03-31
>
> ## Author Response to Reviewer gWzA
> We sincerely thank Reviewer gWzA for the recognition of our work. The further questions are addressed as follows.
>
> ### Q1: Clear definition of ODE distillation
> Section 2.3 of the original paper gives the general definition, and Eq. (3) specifies the concrete ODE distillation used in Self Forcing. Following your suggestion, we will restate this definition in the section discussing how frame-level injectivity should be enforced.
>
>
> ### Q2: Simplified implementation of frame-wise injectivity
> Thank you for the comment. The theory looks more complex because it explains why Self Forcing’s ODE initialization is theoretically invalid, and this setting is itself inherently more complicated. By contrast, the implementation in Eq. (8) is simple because once the teacher is changed to an AR diffusion model, frame-level injectivity is naturally satisfied: each frame is uniquely determined by the causal history, and the invertibility of the PF-ODE is a standard property in the diffusion literature. We will clarify this more clearly in the revised version.
>
>
> ### Q3: Clarification on Fig. 5
> We thank the reviewer, but we believe this is a potential misunderstanding. The “blurry background” in Fig. 5 is actually the mist of the waterfall. In more demos in Fig. 6, the backgrounds of videos generated by our model are not blurry, so this is not a model quality issue. We will clarify this point in the revised version.
>
>
> ### Q4: More video comparisons and why our method improves motion dynamics
> - We sincerely apologize for not providing videos of the other compared methods in the supplementary material. However, the ICML rebuttal policy does not allow adding content other than figures and tables, so we plan to update the open-source project page after the review process to include these videos.
> - We generally attribute the improvement in dynamics to the benefit of resolving the architectural gap, which surpasses both CausVid and Self Forcing. The specific mechanism can be explored in future work.
>
>
> ### Q5: Long-video performance
> Building on video length extraplation methods which are orthogonal to our approach, Causal Forcing can naturally scale to 30-second generation (6$\times$ the training length) without quality degradation. For example, when built on top of LongLive, it achieves a quality score on the 30-second VBench-Long benchmark that is comparable to its 5-second performance (**84.47 vs. 84.59**). It also significantly outperforms Self Forcing-based LongLive (**83.09**). If more advanced video length extrapolation methods become available in the future, we can extend our model to longer video generation.
>
> |Method|Causal Forcing (5s)|Causal-Forcing-based LongLive (30s)|Self-Forcing-based LongLive (30s)|
> |---|---:|---:|---:|
> | Quality Score $\uparrow$ |84.59|84.47|83.09|
> ### Q6: Limitation and future works
> We believe an important challenge is long-term memory. Although Causal Forcing improves the quality of long-video methods, spatial memory is still not fully resolved and remains an open problem.
>
>
> ### Q7: How to train a good Causal Forcing method
> Scaling up the data quantity and improving the quality can further improve performance, as illustrated in our response to RUam Q3. In addition, increasing the training steps in the first two stages and using larger models can also lead to further gains.

---

> > ### Author Rebuttal · Reviewer_gWzA · 2026-04-02
> >
> > The rebuttal adequately addresses most of my concerns. Regarding the improvement in dynamics, the rebuttal does not fully address that. It will be good to provide some further insights on that. Overall, this is solid work and make good contributions to the field. I will keep my positive rating, which is already high.

---

> > > ### Author Response · Authors · 2026-04-02
> > >
> > > Thank you very much for the recognition of our work and the positive feedback on our response. We sincerely appreciate your time and consideration.

---

### Official Review · Reviewer_RUam · 2026-03-12

**Soundness:** 3
**Presentation:** 4
**Significance:** 3
**Originality:** 3
**Overall Recommendation:** 5
**Confidence:** 4

**Summary:**

The paper identifies a flaw in distilling bidirectional video diffusion models into autoregressive (AR) few-step generators. ODE distillation requires injectivity: each noisy input must map to one clean output. For bidirectional models, injectivity holds at the video level but breaks at the frame level because the same noisy frame can produce different clean frames depending on the other frames in the sequence. An AR student learning this non-injective mapping collapses to a conditional mean and produces blurry output. The authors prove this formally and validate it with a controlled experiment. Their solution, Causal Forcing, trains an AR diffusion model via teacher forcing first, then uses it as the teacher for ODE distillation. They also show that teacher forcing outperforms diffusion forcing for AR training due to a train-inference distribution mismatch. On VBench with Wan2.1-1.3B: +19.3% Dynamic Degree, +8.7% VisionReward, +16.7% Instruction Following over Self Forcing, at the same inference speed (17.0 FPS).

**Compliance With Llm Reviewing Policy:**

Affirmed.

**Key Questions For Authors:**

1. Please report total wall-clock GPU-hours for the full Causal Forcing pipeline (AR teacher training + ODE distillation + DMD) versus Self Forcing's full pipeline on the same hardware. If total cost is substantially higher, the "same budget" framing needs revision.

2. Can you run on at least one additional base model (Wan2.1-14B or a different backbone)? Even partial results would strengthen generality. Single-model validation is the main factor preventing a strong accept.

3. Are there failure modes or prompt categories where Causal Forcing does not outperform Self Forcing? A per-category VBench breakdown would be informative.

**Limitations:**

Partially. The authors note the rudimentary CD extension and single base model. hey do not discuss total pipeline compute cost including AR teacher training, dependence on synthetic training data quality, or failure modes.

**Strengths And Weaknesses:**

Strengths:

- The frame-level injectivity observation is a genuine conceptual contribution. Most practitioners would assume bidirectional-to-AR distillation should work if bidirectional-to-bidirectional does. The paper identifies precisely why it doesn't, and the decomposition from video-level to frame-level injectivity had not been formalized before.
- The controlled experiment in Figure 2 is well-designed. Initializing the AR student with standard DMD removes the sampling-step gap while keeping the architectural gap. Performance still degrades, isolating the problem to ODE initialization.
- Table 2's ablation is convincing. The frame-wise comparison is striking: Self Forcing ODE + DMD gets Dynamic Degree 2 and VisionReward 1.951, while Causal ODE + DMD gets Dynamic Degree 64 and VisionReward 6.204. A 32x improvement from ODE initialization alone, exactly as the theory predicts.
- The teacher forcing vs. diffusion forcing finding (Proposition 3.4) is contrarian and well-supported. Diffusion forcing conditions on noisy history during training but clean history at inference, creating a mismatch that causes collapse (Figure 4).
- Fair comparison: same base model, same distillation budget, same inference throughput and latency. All gains come from better initialization, not more compute at inference.
- Figure 3 clearly illustrates the three distillation settings and why the bidirectional-teacher/AR-student case breaks injectivity. Theory and experiments reinforce each other throughout.

Weaknesses:

- Only one base model (Wan2.1-1.3B). The theory should apply broadly, but single-model validation is a limitation for a paper claiming a fundamental principle.
- The total compute comparison may be misleading. Causal Forcing requires training an AR teacher (2K steps on 3K synthetic videos) before ODE distillation. Full pipeline GPU-hours are not reported.
- The 3K synthetic video dataset for AR teacher training is a dependency the paper does not ablate. Quality and diversity of these videos directly affect the teacher.
- No failure mode analysis or per-category VBench breakdown.
- User study is small (10 participants, 10 prompts).

---

> ### Author Rebuttal · Authors · 2026-03-31
>
> ## Author Response to Reviewer RUam
> We sincerely thank Reviewer RUam for the recognition of our work. The further questions are addressed as follows.
> ### Q1: Additional base model
> As suggested, we add compraion betwwen Self Forcing and Causal Forcing on **Wan2.2-5B**. As illustrated in the following table, our Causal Forcing still consistently outperforms Self Forcing across all metrics. The strong result of this additional model further validate the effectiveness of our method.
>
> | Method | Total ↑ | Quality ↑ | Semantic ↑ | Dynamic. ↑ | Vision. ↑ | Instruct. ↑ |
> |---|---:|---:|---:|---:|---:|---:|
> | Self Forcing | 82.33  | 82.91 | 80.03 | 21 | 3.102 | 41 |
> | **Causal Forcing** | **84.44** | **85.07** | **81.92**| **62** | **6.531** | **59** |
>
> The experimental setup is the same as in the main text, except for a batch size of 32, due to limited time and computational resources.
>
> ### Q2: Wall-clock GPU-hours comparison
> As suggested, we add GPU hours on A800 in the following table. Causal Forcing has a training time roughly comparable to the original Self Forcing setup, at about 110% to 120% of its cost. We will include the table in the revision and replace "same budget" with a more modest wording.
>
> | Method | TF AR Diffusion Data | ODE Paired Data | TF AR Diffusion | ODE Initialization | DMD | Total |
> |---|---|---|---:|---:|---:|---:|
> | Self Forcing | - | ODE data (16K, 800 GPU·hours) | - | 3K steps, 560 | 213 GPU·hours| 1573 GPU·hours|
> | Causal Forcing | $\mathcal{D}_{Bi}$ (3K, 150 GPU·hours) | $\mathcal{D}_{Causal}$ (3K, 383 GPU·hours) | 2K steps, 729 GPU·hours| 1K steps, 364 GPU·hours| 266 GPU·hours| 1892 GPU·hours (~120% of SF) |
> | Causal Forcing w/o $\mathcal{D}_{Bi}$ generation | real data (0 GPU·hours) | $\mathcal{D}_{Causal}$ (3K, 383 GPU·hours) | 2K steps, 729 GPU·hours| 1K steps, 364 GPU·hours| 266 GPU·hours| 1742 GPU·hours(~110% of SF) |
>
> Notably, **$\mathcal{D}_{Bi}$ is optional and can be replaced directly with real data**, thereby removing its generation cost. As noted in the following response Q3，this alternative setting even yields better results. Overall, the total cost of the two methods is comparable.
>
>
> ### Q3: Dataset ablation
> As suggested, we add a data ablation study. Causal Forcing trained on different datasets consistently delivers strong performance. Using higher quality synthesized data, or directly using real data, can further improve performance. The results are illustrated as follows.
> | Data (all are 3K in number) | Total ↑ | Quality ↑ | Semantic ↑ | Dynamic. ↑ | Vision. ↑ | Instruct. ↑ |
> |---|---:|---:|---:|---:|---:|---:|
> | Wan 2.1-1.3B synthesized (used in the paper) | 84.04 | 84.59 | 81.84 | **68** | 6.326 | 56 |
> | Wan 2.2-14B synthesized (higher quality)| **85.20** | **85.92** | **82.31** | 67 | **6.994** | **59** |
> | OpenVid，non-synthetic | 84.26 | 84.88 | 81.78 | 63 | 6.421 | 61 |
>
> ### Q4: Per-category VBench breakdown and failure mode analysis
> We add the VBench sub-metric as illustrated as follows:
>
> | Method | Imaging Quality | Motion Smoothness | Background Consistency | Subject Consistency | Overall Consistency | Temporal Style | Appearance Style | Human Action |
> |---|---:|---:|---:|---:|---:|---:|---:|---:|
> | Self Forcing | 70.31 | 97.07 | **97.16** | 96.81 | 24.69 | 24.68 | 20.01 | 94 |
> | Causal Forcing | **71.75** | **98.79** | 96.35 | **96.99** | **24.78** | **25.69** | **20.47** | **97** |
>
> The failure mode is that Background Consistency is slightly lower. This may be because Self Forcing has lower dynamics (Tab. 1), leading to a more stable background in specific examples. However, Subject Consistency & Overall Consistency are both stronger than Self Forcing, indicating that consistency is comparable，while all other metrics are uniformly stronger，especially VisionReward and user rating in Tab. 1.
>
> ### Q5: User study
> As suggested, we expand the user study to 20 prompts and 50 people, for a total of 1,000 valid samples. The results are illustrated as follows, with Causal Forcing still in first place：
>
> | Model | LTX-1.9B | Wan2.1-1.3B | NOVA | Pyramid Flow | SkyReels-V2-1.3B | MAGI-1-4.5B | CausVid | Self Forcing | Causal Forcing |
> |---|---:|---:|---:|---:|---:|---:|---:|---:|---:|
> | Rating ↓ | 6.301 | 2.489 | 8.458 | 6.210 | 6.387 | 6.596 | 4.219 | 2.619 | **1.721** |

---

> > ### Author Rebuttal · Reviewer_RUam · 2026-04-04
> >
> > The authors have thoroughly addressed all five of my concerns. The GPU-hour breakdown shows the total cost is comparable to Self-Forcing. The dataset ablation and per-category VBench breakdown are both informative additions. I am satisfied with the responses and will maintain my positive score.

---

> > > ### Author Response · Authors · 2026-04-04
> > >
> > > Thank you very much for the recognition of our work and the positive feedback on our response. We sincerely appreciate your time and consideration.

---

### Official Review · Reviewer_S2ED · 2026-03-12

**Soundness:** 3
**Presentation:** 3
**Significance:** 3
**Originality:** 4
**Overall Recommendation:** 5
**Confidence:** 4

**Summary:**

**Task and Motivation**: Current real-time video generation approaches typically distill pretrained bidirectional video diffusion models into few-step autoregressive (AR) students. This paper targets the **frame-level injectivity** issue present in prior pipelines and proposes **Causal Forcing** to mitigate the resulting blurring and temporal inconsistency.

**Methodology**: The authors first train an AR diffusion model via teacher forcing, then use it as the teacher for ODE distillation to initialize the subsequent DMD stage. Theoretical analysis demonstrates that, compared to existing autoregressive video diffusion distillation methods, the proposed Causal Forcing effectively mitigates the architectural gap.

**Compliance With Llm Reviewing Policy:**

Affirmed.

**Final Justification:**

The authors’ rebuttal has addressed most of my major concerns. And other reviewers also lean positive, so I support acceptance of this paper. Overall, I believe the work is solid, relevant, and suitable for presentation at the conference.

**Key Questions For Authors:**

Questions are listed in the weakness part.

**Limitations:**

yes

**Strengths And Weaknesses:**

**Strength and Core Contributions**:

1. The paper formalizes and proves the architecture gap issue in current pipelines and proposes Causal Forcing. The method is grounded in solid theory and demonstrates clear empirical gains, making it likely to be influential for real-time and long-video generation tasks.

2. The manuscript is well organized and logically coherent. The figures and experimental charts effectively convey the framework and results, enhancing understanding and reproducibility.

**Weakness, Questions, and Suggestions**:

1. **Long-video performance**: How long can your approach generate frames continuously without degradation in quality? And are there any limitations on the sequence length?

2. **Human evaluation scale**: The current user study includes only 10 participants and 10 prompts. It is recommended to expand the scope of the human evaluation.

3. **Ablation in Tab.2**: Why does the frame-wise setting underperform the chunk-wise setting? And whether the paper’s concept of frame-level injectivity generalize to chunk-wise or hybrid architectures?

4. **Training and data construction costs**: In Sec. 4.1 the authors claim comparable training budgets with the Self-Forcing baseline. Please explicitly report the time and GPU-hour costs for constructing $\mathcal{D}_{Causal}, \mathcal{D}_{Bi}$ and for training the teacher and student models.

---

> ### Author Rebuttal · Authors · 2026-03-31
>
> ## Author Response to Reviewer S2ED
> We sincerely thank Reviewer S2ED for the recognition of our work. The further questions are addressed as follows.
>
> ### Q1: Long-video performance
>
> Building on video length extraplation methods which are orthogonal to our approach, Causal Forcing can naturally scale to 30-second generation (6$\times$ the training length) without quality degradation. For example, when built on top of LongLive, it achieves a quality score on the 30-second VBench-Long benchmark that is comparable to its 5-second performance (**84.47 vs. 84.59**). It also significantly outperforms Self Forcing-based LongLive (**83.09**). If more advanced video length extrapolation methods become available in the future, we can extend our model to longer video generation.
>
> |Method|Causal Forcing (5s)\||Causal-Forcing-based LongLive (30s)\||Self-Forcing-based LongLive (30s)\||
> |---|---:|---:|---:|
> | Quality Score ↑ |84.59\||84.47\||83.09\||
>
> ### Q2: Human evaluation scale
>
> Thank you for the comment. We expanded to 20 prompts and 50 people, for a total of 1,000 valid samples. The results are illustrated as follows, with Causal Forcing still in first place：
>
> | Model | LTX-1.9B | Wan2.1-1.3B | NOVA | Pyramid Flow | SkyReels-V2-1.3B | MAGI-1-4.5B | CausVid | Self Forcing | Causal Forcing |
> |---|---:|---:|---:|---:|---:|---:|---:|---:|---:|
> | Rating ↓ | 6.301 | 2.489 | 8.458 | 6.210 | 6.387 | 6.596 | 4.219 | 2.619 | **1.721** |
>
> ### Q3: Ablation in Tab.2
> Compared with the frame-wise setting, the chunk-wise setting has a smaller gap from the bidirectional model, and is therefore easier to adapt, making it better. In addition, frame-level injectivity also applies to the chunk-wise model: one only needs to treat a chunk as a “frame”, i.e., chunk-wise injectivity, and the formula remains unchanged. We will add this in the revised paper.
>
> ### Q4: Training and data construction costs
> As suggested, we add GPU hours on A800 in the following table. Causal Forcing has a training time roughly comparable to the original Self Forcing setup, at about 110% to 120% of its cost. We will include the table in the revision and replace "same budget" with a more modest wording.
>
> | Method | TF AR Diffusion Data | ODE Paired Data | TF AR Diffusion | ODE Initialization | DMD | Total |
> |---|---|---|---:|---:|---:|---:|
> | Self Forcing | - | ODE data (16K, 800 GPU·hours) | - | 3K steps, 560 | 213 GPU·hours| 1573 GPU·hours|
> | Causal Forcing | $\mathcal{D}_{Bi}$ (3K, 150 GPU·hours) | $\mathcal{D}_{Causal}$ (3K, 383 GPU·hours) | 2K steps, 729 GPU·hours| 1K steps, 364 GPU·hours| 266 GPU·hours| 1892 GPU·hours (~120% of SF) |
> | Causal Forcing w/o $\mathcal{D}_{Bi}$ generation | real data (0 GPU·hours) | $\mathcal{D}_{Causal}$ (3K, 383 GPU·hours) | 2K steps, 729 GPU·hours| 1K steps, 364 GPU·hours| 266 GPU·hours| 1742 GPU·hours(~110% of SF) |
>
> Notably, **$ \mathcal{D}_{Bi} $ is optional and can be replaced directly with real data**, thereby removing its generation cost. As noted in the response to Reviewer RUam (Q3)，this alternative setting even yields better results. Overall, the total cost of the two methods is comparable.

---

> > ### Author Rebuttal · Reviewer_S2ED · 2026-04-02
> >
> > I sincerely thank the author for responding to my questions and for making significant additional contributions to the paper. I believe this is a really solid work and will maintain my strongly positive score. Thanks.

---

> > > ### Author Response · Authors · 2026-04-02
> > >
> > > Thank you very much for the recognition of our work and the positive feedback on our response. We sincerely appreciate your time and consideration.

---

### Decision · Program_Chairs · 2026-04-30

**Decision:**

Accept (regular)

**Comment:**

Strengths: The reviewers unanimously praised the paper's core contributions, particularly the formalization of frame-level injectivity and the robust empirical results.

Rebuttal: All four reviewers found their concerns adequately resolved. The only remaining minor critique is a request for deeper insights into exactly why the motion dynamics improved so significantly, which the authors should address in the camera-ready version.

Given the strong reviewer consensus and effective rebuttal, the AC agrees with the favorable ratings and recommends acceptance.